# Heterogeneous Network Architecture for Integration of AI and Quantum Optics by Means of Multiple-Valued Logic

**Alexey Yu. Bykovsky**

P.N. Lebedev Physical Institute RAS, Leninskiy pr. 53, 119991 Moscow, Russia; bykovskiyay@lebedev.ru

**Abstract:** Quantum optics is regarded as the acknowledged method to provide network quantum keys distribution and in the future secure distributed quantum computing, but it should also provide cryptography protection for mobile robots and the Internet of Things (IoT). This task requires the design of new secret coding schemes, which can be also based on multiple-valued logic (MVL). However, this very specific logic model reveals new possibilities for the hierarchical data clustering of arbitrary data sets. The minimization of multiple-valued logic functions is proposed for the analysis of aggregated objects, which is possible for an arbitrary number of variables. In order to use all the useful properties of the multiple-valued logic, the heterogeneous network architecture is proposed, which includes three allocated levels of artificial intelligence (AI) logic modeling for discrete multiple-valued logic, Boolean logic, and fuzzy logic. Multiple-valued logic is regarded as the possible platform for additional secret coding, data aggregation, and communications, which are provided by the united high dimensional space for network addressing and the targeted control of robotic devices. Models of Boolean and fuzzy logic are regarded as separate logic levels in order to simplify the integration of various algorithms and provide control of additional data protection means for robotic agents.

**Keywords:** multiple valued logic; Allen–Givone algebra; quantum network; multiagent system learning; fuzzy logic; heterogeneous architecture

## 1. Introduction

### 1.1. Actual Problems for Communication Networks

The design of new generations of Internet [1–6] and quantum networks [7–9] can be regarded as the new step in the long history of communication networks (CN) [10,11]. CN are sets of nodes, see Figure 1, where any pair of nodes A,B,... can exchange useful data via a set of communication lines, connecting some of the nodes. The goal is to realize the data transfer for an arbitrary pair of nodes and to provide the optimal choice of the route for the data transmission. The most known types of CN are shown in Table 1, and demonstrate the wide spectrum of possible schemes. Note that in elder generations of CN, the term "node" mainly means the combination of a signal procession device and a qualified person, who makes a decision and uses the delivered message. However, modern people want robots to make decisions even in critical tasks, preferring to exploit CN for education, social activity and entertainment. Thus, the term "computer network" now defines the network of computing devices, which can include robots [12–14] and sensor networks [15–18]. In any way, the desire to achieve the new level of artificial intelligence (AI) for future CN and global 5G and 6G Internet is partially motivated by the need to provide high throughput and reliable content delivery both for traditional network nodes [19–21] and unmanned robotic systems [22–26].

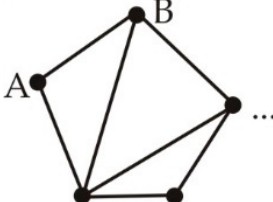

**Figure 1.** A communication network is regarded as a system of transmitting nodes (A, B,...) connected by data transmission lines.

**Table 1.** The prehistory of modern communication networks (dates are compiled from [10,11]).

| Industrial and intrafacility networks for control and management (1890s–1900s) | Network of traffic lights (1914) | Radioloca-tion network (1940s) | Satellite communi-cations (1958) | Mobile telephones (1973) | Fiber optics (1975–1980) |
|---|---|---|---|---|---|
| Internet (1983) | Project of distributed quantum computing (1990s) | Project of network-centric system (1999) | Project of photonic network (2000) | Multiagent system of robots (2000s) | Quantum key distribu-tion (2005) |

Actual application fields for AI methods are given in Table 2. As a whole, they are aimed at the imitation of facilities of qualified people in software and hardware agents. Within the framework of the given paper, the actual AI methods [27] can be mainly subdivided into multiagent systems (MAS), neural networks, and fuzzy logic systems. MAS are primarily applied for the modeling of software and hardware agents in the tasks of interaction with the environment and other agents, as well as for space positioning [28–35]. Modern versions of deep learning neural networks are concentrated on computer vision and speech procession [36–39]. Fuzzy logic has proved to be the effective method to work with uncertain and approximate data, substantially reducing the volume of calculations for autonomous vehicles, industrial systems, and medicine images processing [40–43]. Fuzzy systems can also be integrated with neural networks in neuro-fuzzy systems [43]. Such interesting and actual items of AI research, as reasoning, planning, and data representation methods [27] are out of the frame of the given paper, but in Table 2, they can be associated not only with industrial robots [33,34] and unmanned transport [23–25], but also with data leakage protection [9] in computer security systems, where the task is to monitor the illegal activity of the staff. Quantum key distribution (QKD) is here the method of confidential generation of cryptographic keys for CN [8,9], as quantum computing is the prospective way to raise the computing performance for some classes of network tasks and for quantum schemes of MAS learning and neural networks [44–46].

Thus, all these trends of CN are mainly not competitors, but rather the complementary AI toolkit for producing intellectual tasks at distant network nodes. That is why it is necessary to provide synergy and to design new ways for their efficient joint applications.

**Table 2.** Actual application fields for modern types of communication networks (CN) [1–6,15,16,18,22,23,25,26,43,44]. IoT: Internet of Things.

| Computing | Defense | Computer Security | Social Security | Finance and Commerce | Search and Storage of Data |
|---|---|---|---|---|---|
| Cloud, Supercomputer, Cryptocurrency mining, Quantum computer | Global systems for control, detection, and monitoring, Robotics swarms | Data leakage protection, Computer monitoring of staff activity, Quantum key distribution | Face control, Biometrics, Search of illegal cargo | Accounting, Billing, Document exchange, Advertising, Marketing E-commerce, | Data and knowledge search, Expert systems, Machine text and speech translation |
| **Medicine** | **Computer Vision** | **Sensor Networks** | **Industrial Robotics** | **Navigation and Space Positioning** | **Unmanned Vehicles and Logistics** |
| 3D tomo-graphy, Distant consulting, Health control and surgery, Robotic nursing | Image and scene processing for defense, transport, and industry | Objects detection, Positioning control | Control of semi- and fully unmanned systems | Optimal routing, Multiagent mapping, | Optimal logistics, Unmanned traffic control, Automatic repair and service |
| **Education** | **Mass Communications** | **Mass Media** | **Entertainment and Travel Industry** | **Social Activity and Networks** | **Household** |
| E-Libraries, Data search, Educational films andsimulators entry | Mobile and stationary telephone, Skype, Internet | Delivery of news and targeted packets | Search of network content, Hotel and ticket reservation | Polling, Computer voting, Social projects | IoT, Food ordering Automatic repair services |

*1.2. Quantum Data Protection Methods for CN*

Modern versions of quantum CN, see e.g., recent reviews [7–9], solve the task of quantum key distribution (QKD), i.e., the trusted delivery of secret random keys to subscribers, and are expected to protect distributed quantum computing in the future. The mass implementation of the QKD method for CN is not as rapid as was expected in the 2000s [8,9]. It faces a number of problems, which are not solved yet. Modern quantum memory schemes are not ready for practical use, the length and the data rate for QKD are limited, and the problem of quantum and classical attacks is not solved at the appropriate level [8,9]. Nevertheless, modern quantum cryptography schemes seem to form very significant additional barriers for malicious users, as now they should combine classical hacker tools with the development of specific quantum optics equipment and algorithms, which complicates their activity and simplifies their detection.

One of the substantial unsolved problems is that quantum cryptography uses many auxilliary perconal computers (PCs) and field-programmable gate arrays (FPGAs), which should exploit traditional antiviruses, firewalls, antispam software, means of counteracting false flow attacks, different means to protect against unauthorized access, and physical hacking [9]. However, the modern trend for all types of CN is that alongside encryption methods, they should exclude data leakage and unloyal staff activity, which requires the intensive research of AI methods for the aggregation and analysis of characteristic features of different documents and events. In a general case, the analysis of data leakages is supposed to include multi-parametrical processing of rare and not regular events, which should differ somehow from regular data traffic, which is typical for the legal labor activity of an eavesdropper. In spite of that fact, that quantum schemes for mobile robotics were firstly designed for aircrafts and cosmic systems [47,48], and even for these expensive and usually well-protected systems, the data leakage problem can be substantial. For cheap and simple industrial robots, this problem is much more difficult to be solved. Thus, on one hand, the understanding of the scene by the robot is the priority task for the further successful integration of MAS into CN [27], but on the other hand, it is the component for quantum network security. Here, computer vision should be regarded as one of critical components that is necessary for the security of both quantum and traditional systems.

Another aspect of quantum network security is the problem of seamless computing [49]. Seamless computing can be interpreted as the ability to execute individual and joint tasks by means of all types of computing devices, possessing different resources and software versions. However, artificial or natural radio or optical interference can be the obstacle for the correct functioning of both quantum and classical devices, preventing the successful interaction of different types of devices. That is why a high dimension space is needed in CN not only for global addressing, but also for the targeted description of various tasks and models of agent's behavior during their collective work [15,16,18,23,24], supported by reliable communications channels.

In any way, the project of purely photonic networks [50] has led to the design of the special multiagent system (MAS) for the total control of all quantum and traditional cryptography keys. So that CN need AI methods for both quantum and classical subsystems.

*1.3. Achievements in the Realization of AI Methods for CN*

The concept of agents and multiagent systems (MAS) was formed over several decades, and its most comprehensive description was done in [27]. The development of many specific aspects of AI and MAS was disclosed in [37–43]. MAS can be considered as an additional way to organize parallel computing [51], as it includes the division of a complicated task into several more simple ones, done by the group of agents. Agents can be realized as software bots [27,52,53] and as hardware devices [12–14,41,42], for which microprocessor platforms and auxiliary subsystems need specific software for the support of energy systems, numerous sensors, and actuators. The interaction of an agent (or a robot) with other ones and with the environment [27] can be characterized by the set of at least 29 basic features of agents [53], which are necessary for the successful imitation of human behavior. This set includes autonomy, activity, reactivity, communicativeness, goal setting,

and many other properties. Although during two last decades the spectrum of basic features did not changed principally [27,40,51–53], their interpretation can be mixed in different projects. That is why the traditional problem is to provide all these concepts for different platforms, essentially touching upon the question of seamless computing.

The most significant impetus to the development of network robotics was given in the 1999 by the global concept of a network-centric system, which was aimed at defense tasks [54]. It has induced the intensive research of collective multiagent systems for collective mapping and space navigation [12,55], unmanned logistics [31] and swarm robotics [55]. Up to 2030, the substantial part of transport traffic is expected to be served by unmanned systems and guided by the network of radiofrequency beacons [56]. All these projects suppose the flexible routing and distant monitoring of robotic vehicles [19].

However, further advances in mobile robotic systems strongly depend on the progress in the sphere of computer vision. This field of investigations has more than a half a century of history, but neither the development of traditional methods nor the progress in the design of MAS raised the computer vision to the needed level [27,57,58]. This field proved to be very complicated, and now, it is represented mainly by new versions of neural networks and by the large number of fragmented semi-empirical algorithms [57], which have been collected as an open library within the project Visual Studio 2015 [58].

Sensor networks [15–18] should be regarded as a separate network component, partially compensating for the lack of autonomous computer vision systems. Such systems consist of sensor modules equipped with quick and low power consumption controllers that are adapted for long autonomous network data transfer.

Since 2015, the new impetus was given to computer vision design, which was caused by advances in the development of neural network deep learning methods [36–39].

These investigations do not have principal contradictions with concepts of MAS and agents, as they also (see Chapter 6 of [27]) exploit decision tree methods. The difference here is that traditional AI learning procedures [27] are often based on formation of lists of solutions, labeled examples, and reinforcement, which can be interpreted within predicate calculus. However, substantial progress was recently obtained namely in neural network deep learning methods, whose methodology is much more close to the so-called threshold logic, presented earlier in the 1960s [59,60], but not to predicate models. In any way, the concept of agents seems to be flexible and universal enough to integrate all different types of logic models, including multiple-valued ones.

*1.4. Classical Schemes for Secured Robotics*

The specific problems of secure communications in modern versions of MAS were discussed in 2010 in the review [61], where the authors has stressed that every agent should analyze the network parameters with respect to its own machine, but the detection of possible attacks should be processed in a collaborative regime. Here, the initial level of agent-to-agent interaction includes different levels of security solutions, regulated by the National Institute of Standards and Technology (NIST) [62]. It supposes agent-to-platform security tasks based on the control of agents by means of the secure execution of the environment, protecting from malicious agents. The second level of protection supposes platform-to-agent security means, provided by the some kind of privacy computation in agents. It should protect an agent from possible manipulations regarding workflow, privacy, and the integrity of data. The third level of protection includes agent-to-agent security and supposes that an agent tries to protect itself from denial of service, the pretentious actions of other agents, and spying. Moreover, special platform-to-platform security means are designated for securing interactions between different platforms. The authors of [61] have emphasized the role of user-password authentication for agents and the role of trusted and untrusted software objects, which are created by its own and by external servers. Special attention was given in [61] to the estimation of trust and vulnerability factors, which are necessary for authorization and trust management systems. In addition, the role of peer-to-peer networking was shown, which provides strong authentication service, in which

authorization, secure transport, and secure execution are provided for the agents. One should envisage special protection from common computer attacks and infections by malicious software. The review [61] was concluded by the recommendation to develop two types of solutions: secure MAS for the design of general purpose applications, and security MAS, which provides a new overlay for distributed security models of MAS.

Five years later, in [63], the authors have noted that a new paradigm has been created, which was named agent-oriented programming and was based on concepts of AI in distributed systems. Primarily, attention was given to models of an agent's role and communications. For the agent level, new designs included enhanced identification and authentication steps, which include special codes for the host name, agent's name, and permissions, which are used in the header of a robotic message. At the system level, the gate agent was discussed as responsible for (1) handling all communications between the main host system and other components, (2) monitoring the communication flows, (3) breaking the communication channels in case of emergency, and (4) keeping mobile agents out of the main host system.

In [63], the threats at the system level were classified as coming from:

- mobile agents to hosts,
- the Internet (denial of service, damage, event triggered, compound, and user attacks),
- altering the logging system and agent code, data, and configuration,
- fake agent and fake service.

For the agent level, the threats were subdivided [63] as coming:

- from hosts to agents,
- from agents to agents,
- from users to agents,
- for communication among agents, including identification and authentication, unauthorized access, message injection, knowledge injection, and other malicious impacts.

In 2019, the paper [64], which was presented at the Autonomous Agents and Multiagent Systems (AAMAS) conference, has demonstrated the trend to design trusted AI and trust modeling. These aims were formulated to obtain fairness, transparency, dangers, and collaboration for MAS systems. In particular, transparency was associated with learning and deep learning systems. It was interpreted as human interpretability—an explanation of ontologies and recommendations—as well as clear interpretability for given advices. Thus, MAS agents should receive new substantial features of acceptance, which are peculiar to human beings. Another interesting trend is the design of trust models for computer vision systems, which include the control of training and testing sets as initial data for the formation of the trust model. In addition, monitoring of the reputation model for an agent has become an actual task, which differs for stages of training and the performance of the vision system. Thus, reasoning systems [64] have become now much more actual for trust and reputation control.

Another aspect of MAS reasoning is the design of a special language for robotics communications in multiagent systems. The most known robotics languages were represented by NesC and Tiny DB [65,66], which were well adapted for distant control. For future generations of networks, the actual task is to design autonomous agents that do not need an intensive transfer of instructions. Only if the robot can not carry out the task based on his own knowledge base should other collaborating robots or the external monitoring system help and provide him with absent knowledge. That is why the agent model should envisage a special structure of parameters for collective learning and the external correction of its rules and data sets [14]. The modern trend for IoT, autonomous transport, and industrial robotics is oriented to provide the full range description of targeted tasks, beliefs, and desires [14,16,24–27], which makes the design of the universal language structure for typical AI tasks quite actual.

However, the modern level of data protection for transport and logistics agents is based mainly on traditional cryptography protocols for Internet and mobile communications [56]. The author of the paper [56] has analyzed communications for levels vehicle-to-vehicle (V2V) and vehicle-to-infrastructure (V2I). An actual problem was formulated as the creation of a flexible structure of cryptography keys, responding to the growing demands of European transport infrastructure. Main attention was given to the design of complex structure of long and short-range certificates of authorization, which were transmitted between vehicles and traffic control stations. The designed in the system of [56] was widely spread in Europe protocol IEEE 802.11p, which was based on the calculation of elliptic curves according to standard NIST P-256. However, the authors have mentioned that such a cryptography system can not protect from specific types of side channels attacks, and new methods are needed here. Moreover, it was discussed in [9] that elliptic curves-based cryptography was declared by NIST to have principal vulnerabilities.

*1.5. MVL-Based Schemes for Secured Coding of Data*

The problem of direct secured communication of agents by means of MVL protocol was discussed in [67,68], and the proposed MVL version of the one-time pad (OTP) secret coding scheme [67] was initially oriented on the application of quantum random number generators [69,70] for the protection of MAS systems. Since then, the development of quantum key distribution networks [8,9] has greatly escalated the interest in OTP secret coding protocols. As it was discussed in [9], an MVL-based version of OTP secret coding [67] can be used for adverse environmental conditions, preventing the transfer of qubits.

This method even for the 8-bit platform allows one to increase the dimensions of the space of random one-time keys up to $10^{500}$ or more. That is why it can be used for the secure logic control of large-scale robotic and multiagent systems. The increase in the size of the key space is aimed here at counteracting brute-force attacks [8], which are especially dangerous if an eavesdropper uses a cloud service or a quantum computer.

The MVL method is based on the *k*-valued Allen–Givone algebra (AGA) [71], where input and output variables in multiple-valued logic functions take discrete truth values {0, 1, 2, ... , *k* − 1} [72]. The detailed basic technology of AGA functions $y = F (x_1, ... , x_n)$, formation, and calculation is described in Section 2. Given the values of *k* and *n*, the logic function can be calculated by a rigidly defined algorithm, which is appropriate for parallel processing and for the use of FPGAs.

Mass cryptographic techniques usually exploit key-space dimensions up to ~$10^{30}$, but modern mathematics provides much larger dimensions [73,74] for addressing and targeted parameters description. The toolkit of multiple-valued logics [67,68,75] is convenient for implementing high-security one-time pad techniques, and it potentially provides ~$10^{70}$ different random one-time keys even for an 8-bit platform without overwriting the memory of the encoding module, which can be used for long-term autonomous work. The gain in dimension in the MVL method in comparison with binary logic is because for a multiple-valued logic function, the number of rows in its truth table is $k^n$ instead of $2^n$ for Boolean logic [72], and the number of different logic functions is $k^{k^n}$ instead of $2^{2^n}$.

To encrypt messages by MVL, one should install separate quantum random number generator (QRNGs) in encoders and decoders [69,70]. In addition, it is necessary to generate confidentially one or several secret multiple-valued logic functions with randomly given parameters with the help of QRNGs [67,68,75,76] and write it in the memory of both subscribers. In order to construct such a function, it is sufficient to generate several arrays of randomly assigned *k*-digit numbers, and then, according to certain rules, to form a special matrix of randomly chosen parameter pairs $(a_i, b_j)$.

The method of formation and application of random one-time keys in the MVL method is described in detail in [67,75,76]. At the beginning of the secure communication session, abonent Alice, with the help of her QRNG, should generate a one-time key-prompt, i.e., a sequence of $n - 1$ randomly assigned *k*-digit numbers. Alice substitutes this data set as input variables $x_2, ... , x_n$ into a logic expression for the secret function. Then, successively substituting the values of $x_1$ from 0 to $k - 1$, it calculates a

sequence of $k$ random values of the output variable $y = f(x_1, \dots, x_n)$. A one-time key $R$ is a random permutation of a fixed initial sequence $R_0 = \{0, 1, \dots, k-1\}$ produced in the cells of the memory chip with the help of a random sequence $f(0, 0, \dots, 0), \dots, f(0, 0, \dots, k-1)$.

In addition to the original way to realize high dimen sions of random key space [67], the MVL method provides new versions of position-based cryptography [77], exploiting the protocol, designed by Unruh [78]. Position-based cryptography concepts [77] suppose that a robotic agent can independently verify and interpret the space location and other scene parameters of the partner abonent.

The secured transmission of fuzzy rule, linguistic variables, and membership functions was also adapted for MVL secret coding [79]. This algorithm was proposed for controlled ordering/disordering of the structure of fuzzy logic knowledge base, providing remote switching off/switching on of robotic agents. Since the set of fuzzy logic rules 'if ..., then . . . ' and associated membership functions, which are used by the robot's control system, can represent a valuable knowledge structure, it potentially can be stolen by eavesdroppers in the off mode of the robot. Thus, it is necessary to encrypt the knowledge structure for secure storage. Such an MVL algorithm can be implemented in a digital 'key' and a digital 'lock' installed on the robot. The 'lock' regulates the access of its internal subsystems to the recoding tables stored in the random access memory (RAM). With the help of several multiple-valued logic functions and auxiliary sequences of random numbers, the 'key' allows one to remotely erase and restore coding sequences.

One more MVL scheme was proposed in [76] for the verification of data integrity in simple robotic systems. This scheme supposes the coordinated work of two modules in the receiving agent. The packet, transmitted by the first agent, includes three parts, which contain (a) the standard set of Transmission Control Protocol/Internet Protocol (TCP/IP) data, (b) informative data, which can be secretly coded by standard methods, and (c) the repeated informative part of (b), coded by the MVL method [67]. The first receiving module is to decode and to compare parts (b) and (c) in order to check the data integrity. After this, the first module should transfer the verified set of bytes to the second module of the receiving agent, excluding the transfer of any alien data structures. After the session, the first receiving module should be reloaded from a read-only memory (ROM) device. Thus, the malware code infection can be minimized [76] by decreasing the uncontrolled traffic transmitted via traditional network protocols.

The problem of high-dimensional space for seamless addressing in global networks was earlier discussed in [76,80], which was initiated by the design of the special language for parallel computing of tree data structures (abreviated as PARSEK in russian language). The network structure of AGA mapping switching functions [76] was proposed to form high-dimensional tree structures, which are close to simple variants of graph structures. It was shown that AGA functions can provide exclusively high dimensions of the multi-parametrical space. The tree structure for seamless addressing and computing in the global network [76] should use the minimal set of instructions in order to provide an asynchronous regime of work for a distant mobile agent, which can hardly support distant clocking and evaluate time delays for received instructions.

The MVL model of an agent was proposed in [68] and based on the known homeostat model [81], which was aimed at the realization of secured communications by means of the non-alphabet language for an MAS. This homeostat model can be considered as a discrete analog of the fuzzy controller, providing digital outputs for the self-sustaining of several critical parameters within preliminary chosen bands of values. The general scheme of such agent can be characterized as a MVL homeostat controller, scheduling the network structure of several MVL mapping functions [76]. The homeostat's model supposed the use of the working cycle, including the polling of critical sensors parameters, communication modules, and secret coding/decoding modules. The working cycle was ended by the decision making, based on all the obtained information. This scheme needs the formation of time diagrams for the control of all internal and external subsystems, which are switched by MVL functions.

In [82], the heterogeneous MVL model was proposed for the control of the decision-making procedure in the agent, combining the MVL, fuzzy, and traditional Boolean logic models. The idea

was to integrate many diverse models into the holistic model of robotic agent behavior, combining precise and approximate descriptions for space and time parameters. The problem here is that fuzzy logic with its linguistic variables and fuzzy rules has strict limitations for the profile of membership functions, and known fuzzy controllers [40–42] also exploit limited working bands of parameters, thus decreasing the band of working regimes for actuators. This is not critical for domestic appliances, but it is unacceptable for decision-making algorithms in industrial robots. In order to control and to minimize "voids" in the multi-parametrical space of control parameters, it was proposed to apply scheduling, which was expressed formally by MVL switching functions.

Thus, the design of MVL schemes for data protection was initially aimed at the application of acknowledged methods of AI [27], including multiagent schemes and fuzzy logic, for simple microcontrollers MCS-51. Now, such algorithms can be adapted for FPGAs and quantum computing. However, MVL algorithms are very specific and differ greatly from AI methods. That is why the interaction of MVL and AI algorithms is preferably to be controlled within the special architecture, which is proposed in Section 4.

### 1.6. Actual Methods of Data Clustering

The traditional field of optoelectronic data and signal processing for network robotics includes the problems of automated pattern recognition and search of regularities in data structures [83,84]. This item is called pattern recognition and is based on the classification and machine learning tasks [85–87]. The current situation in the sphere of data aggregation and events analysis can be briefly characterized by referencing reviews [88–90].

In the review [88], published in 2010, the situation in the sphere of biomedical research combining the large number of various processing methods was estimated as daunting due to the diversity of cluster analysis, differing terminologies, goals, and assumptions. That is why different methods were compared in detail in [88] according to the mathematical distance and similarity functions, including Minkovsky, Euclidian, squared Mahalanobis, Pearson, point symmetry, and many other metrics, e.g., [91]. First of all, hierarchical and agglomerative clustering algorithms were analyzed. Squared error-based clustering and the problem of a combinatorial burst of possible number of partitions was discussed, where the K-means algorithm was acknowledged to be the most popular method. The exceptional role of approximate fuzzy clustering (realized by Bezdek) was dedicated. In addition, neural network clustering was related to the concept of competitive learning.

The review [89], presented five years later, has characterized the situation as more positive, but it has emphasized that every clustering algorithm has its own strengths and weaknesses, due to the complexity of the information to be processed. It is substantial that the clustering algorithm based on quantum theory was also discussed in [92–94].

Furthermore, the review [90] has represented the interesting problem of mixed data clustering, which deals with both numerical and categorical features. That task is especially actual for such domains as healthcare, finance, and marketing, but this task was called challenging, because it is a doubtful way to apply mathematical operations, such as the summation or averaging for datasets in these domains. Special taxonomy was proposed for the study of mixed data clustering algorithms, which among other methods used hierarchical data structures. To some extent, methods of mixed data clustering correlate with the method of linguistic variables in fuzzy logic were proposed by Zadeh in [94], where a linguistic variable can have the set of several linguistic values, defined by membership functions with special triangle or sigmoid profile. As a whole, mixed data clustering [90] seems to be the new set of solutions for the old problem to imitate human reasoning and operate quickly with compact data structures in the high-dimensional space of symbolic and number values. Here, fuzzy logic provides an adequate description [94] for approximate reasoning, and as it is discussed further, multiple-valued logic [71] represents the correct and precise description for large-scale sets. One should note that both concepts are the consequences of human intellect and are to be used together in AI systems.

The classification task can be subdivided [57,85] into vector optimization methods (regression analysis and support vector machine) and statistical methods (discrete and cluster analysis). All these methods somehow determine the appropriate boundaries between classes. In particular, linear vector optimization methods calculate the linear separating hypersurface, which is visually shown for the 2D case of a measurement grid (see Figure 2a). Typical methods here are to find the linear geometry parameters of the optimal surface separating classes of objects and to maximize the margin distance 2d between classes (see Figure 2b). This task involves the choice of the minimal number of trouble points for the calculation of so-called support vectors [85]. Another popular method is to reduce the nonlinear tasks to linear separability by means of kernel functions [83].

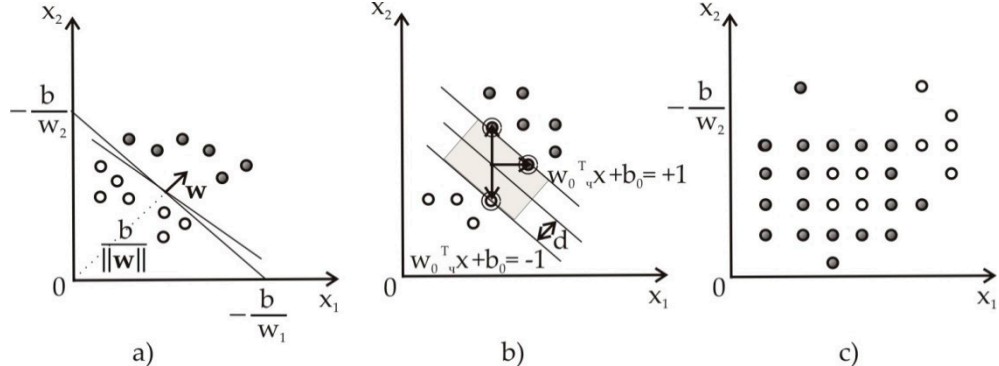

**Figure 2.** (**a**) 2D example of two classes of objects (white and gray), which can be linearly separated by the support vector machine (SVM) method, using the calculation of geometrical parameters for the optimal separating surface. (**b**) The basic idea of the well-known SVM method is to determine the maximal margin 2D length by the calculation of support vectors for the minimal number of objects taken near the border. (**c**) The example of linearly nonseparable sets of white and gray objects.

However, for many situations such as that shown in Figure 2c, it is impossible to find a linear separable surface. That is why fuzzy logic with its uncertain and approximate data processing methods also can be used [40–43]. However, these methods can not provide the correct method to formalize an arbitrary boundary between features classes. The visual example of nonseparable classes is given in Figure 2c and demonstrates the need for the mathematical model, which provides a correct formal description for the arbitrary distribution of classified objects near the margin of two classes.

The spectrum of possible solutions for hardly separable sets includes, e.g., the method to add the calibrated noise component to data; this concept was discussed in the paper [35], which was devoted to the problem of events aggregation in large-scale IoT systems. However, the guaranteed mathematically correct method of arbitrary data separation is proposed in Section 3 to be based on MVL function.

### 1.7. Fuzzy Logic and Approximate Clustering

One more aspect of robotic systems refers to the problem of movement control and space positioning of agents and their executive devices [27,29–31]. Due to adverse external conditions and the possible errors of sensors and actuators, the internal model of the scene in the agent may be inaccurate and incomplete. Such problems are traditionally in the competence of fuzzy logic models, which have been intensively investigated during last two decades [40–42,52,53]. Such systems are based on fuzzy reasoning, fuzzy rules, linguistic variables, and membership functions [91]. The research of fuzzy methods was initiated by Zadeh in the 1960s and has created many successful applications in the sphere of transport, industrial systems, and medicine [29,33,95]. Now, the role of approximate reasoning and fuzzy logic models in AI modeling is too great for modern robotic systems, and many multiagent software and robotic schemes need to include uncertain data processing, fuzzy logic, and approximate clustering into agent models [12–14,52,53]. That is why the desire to obtain a seamless space for robotic decision making leads to the recognition of the need to include fuzzy logic into

the set of basic AI components, which are conjugated with computer vision [57,58]. As a whole, the processing of approximate space and time data is substantial for global and local positioning, collective mapping, unmanned vehicles routing, unmanned logistics, and verification tasks in computer security [30,31,40–42]. Now, fuzzy logic includes a large number of methods, providing the successful work of many subsystems, reproducing musculoskeletal systems of man and animals. These methods are specific and differ from traditional models based on kinematic equations [41]; that is why such system modeling is usually regarded separately.

Fuzzy clustering methods are described in the large number of papers, where earlier works (published in the 1990s) are represented e.g., by [96–99]. A recent level of fuzzy clustering is disclosed, e.g., in papers and reviews [100–103]. Now, flexible fuzzy clustering methods are being applied and provide decision making [100] in many fields, including finance, the energy sector, medicine, Web classification, healthcare, machine learning, pattern recognition, and time series prediction. Comparative performance analysis for fuzzy clustering on standard data sets was represented in [101], which did not find one best variant that was appropriate for all types of noise and outliers.

Modern investigations of space-division multiplexing elastic optical networks (SDM-EON) [102] are actual for the future enlargement of Internet traffic and were proposed to use dynamic unsupervised fuzzy clustering for more intelligent and effective resource assignment. For different sample scales, these fiber systems can use either the known C-means method or the direct clustering method. However, this method is appropriate for the fiber data transmission line with well investigated physical parameters, but it scarcely can be applied for the general case, when the intellectual robot does not have preliminary information about the system, whose data it should analyze.

Briefly, fuzzy clustering was recently characterized in [103] as an unsupervised learning, which can partition identical data patterns basing on some similarity measure, which increases this similarity within one group and decreases it between different groups of objects. The main advantages of fuzzy clustering refer to its simple and straightforward programming, suitable for very large data sets since its time complexity is O(n). In addition, it can produce very good results for some specific clusters parameters (i.e., for hyperspherically shaped well-separated clusters). Fuzzy clustering methods are robust and have been proven to converge to local optimal solutions. At the same time, fuzzy clustering has some drawbacks, as this technology:

- supposes an a priori known number of clusters to be used,
- is sensitive to the cluster centers' initialization phase, and
- is also sensitive to noise and outliers.

Thus, now there are many fuzzy logic clustering methods [104], but there is no consensus on which methods are more suitable for a given database with specific statistics. Moreover, default configuration is not always accurate, and the random search of parameters proves to give a better alternative.

That is why a complicated agent model for future robotics should envisage a large number of different rules, linguistic variables, and membership functions supported by the large number of auxiliary algorithms, based on traditional Boolean logic, and including both statistic and AI methods. At the same time, all these data structures should be reliably protected from leakage and modifications of data, as numerous rules and membership functions [13,40,41,91] are difficult for verification, and potentially any inaccurate or dishonest staff member can create many problems.

### 1.8. The Aim of the Presented Work

As a result, the general requirements for a network with secure agent communications are as follows.

- The level of security for CN should be increased further not only by the development of various QKD schemes [7–9], but also by AI methods of data leakage protection and by the adaptation of a one-time-pad secret coding for different levels of data processing in the agent [67,75].

As possible data leakages due to malicious staff members create threats both for traditional and quantum network subsystems, AI methods should be especially focused on the search, analysis, and recognition of random and suspicious events, which are described in the multi-parametrical space of characteristic features of objects and events. New data clustering methods are needed to be designed both for precise and approximate models, as well as for categorical and linguistic data structures. Future CN should be enhanced by various authentication and verification schemes [75,77].

- Computer vision, speech, and acoustic signals processing are the critical technologies for the knowledge exchange, monitoring of the scene, and for controlling an agent's motivation, goal setting, planning, evaluation of resources, and objects positioning. That is why such image processing methods as data aggregation, clustering, noise filtering, contour extraction, and the analysis of the knowledge structure are very actual for robotic subsystems in global CN [58].

- Peer-to-peer network segments formed by intellectual agents in global CN should be supported by the design of protocols for directly targeted and problem-oriented communications between agents. The traditional address space for future networks is to be complemented by a multi-parametrical space for the targeted addressing of MASs and for multi-parametrical modeling, which is described by the special language (protocol) for direct peer-to-peer communications. Seamless computing in global CN [65] is to be provided for different hardware platforms, and the integration of robots should not create obstacles for routine Internet protocols.

- Learning methods [27], imitating the behavior of qualified men, are actual for the modeling of an agent. As of now there are no universal methods of learning that are good for all possible tasks; thus, heterogeneous models can be applied to form necessary combinations of algorithms. Methods of learning should support objects and events aggregation [35] for large databases and should provide control of possible voids and data dubbing in an agent's knowledge base.

- Fuzzy logic methods [40–42] of uncertain and approximate data processing are necessary for the simulation of a human's behavior and musculoskeletal systems. Such models should be supplemented by new precise methods for the description of borders between classes of objects.

The goal of the paper is to design new methods of data clustering and supporting network architecture, which:

- enlarge the choice of data clustering and classification methods for various types of data,
- are adapted for the integration of additional means of secret coding for agents and MAS, supplementing QKD and traditional network means of data protection,
- provide new schemes for seamless computing and communication languages for global networks,
- can integrate multiagent concepts, fuzzy models, and other AI methods with quantum schemes in global CN.

The solution of these problems is proposed to be based on multiple-valued logic (MVL) switching functions, which are appropriate means to link different AI methods in the multi-parametrical space of control parameters.

Before the discussion of MVL classification methods, the description of an MVL formal model is given in Section 2.

## 2. Basic Methods of Multiple-Valued Logic for Data Processing

### 2.1. Types of Logic Models and the Specifity of Multiple-Valued Allen–Givone Algebra

The history of multiple-valued logic (MVL) was disclosed in [71] and is long enough, as at the end of 19th century it was not yet obvious that Boolean logic will be supported by effective technical platforms. Now, one can find in literature various kinds of logic models, including Boolean, multiple-valued, continuous, fuzzy, hybrid, and threshold ones [59,60,71,72]. In contrast to Boolean

logic, discrete MVL logic deals with logic functions $y = f(x_1, \ldots, x_n)$, where all $n$ input variables $x_1, \ldots, x_n$ and one output variable $y$ have $k$ discrete truth values, which are taken from the set $\{0, 1, \ldots, k-1\}$. Here, 0 is the absolutely false value and $k-1$ is the absolutely true one. Thus, it differs from Boolean logic, which deals with the set of truth levels $\{0,1\}$.

Briefly, the general idea of discrete multiple-valued logic was to raise the "density" of calculations and to enlarge the load for one logic gate. Separate microelectronic components for MVL systems were designed in the 1970s (see Chapters 1 and 2 of [71]), but the priority development of Boolean systems was based on great achievements in microelectronics. Furthermore, the interest in fuzzy logic [40–42] has prevailed and left MVL technologies in the shadows. As a result, some authors have confused concepts of MVL and fuzzy logic. Nevertheless, the difference between them is in the fact that fuzzy logic uses a continuous description of truth levels and uses logic operators MAXIMUM and MINIMUM [71,72], but it is the so-called Kleene algebra [72,105], which is not a complete set of logic operators. This property means that the model formed by this set of logic operators can represent any arbitrary function. Thus, some truth tables for discrete MVL functions (mainly, with 0s and 1s truth levels) can not be expressed by fuzzy logic. Moreover, fuzzy logic has made impressive advances mainly as a method to process uncertain and approximate data. As fuzzy logic rules 'If... Then...' and membership functions [40–42] can be easily described by MVL expressions for control procedures and secret coding schemes [79,106], the relevant solution seems to use an MVL switching scheme for the activation of specialized fuzzy logic modules. For global networks and pattern recognition tasks, it is also convenient to exploit large capacity discrete models for the switching of fuzzy rules and membership functions. In such systems, an MVL model can be defined for a large number of truth levels $k$, which conveniently overlaps with the number of truth levels in fuzzy logic models, but at the same time, it provides the complete set of logic operators. There are several versions of MVL complete sets and the most well known of them are the Lucasievich, Rosser–Terkett, Post, and Allen–Givone systems [71,72]. They all use MAXIMUM and MINIMUM operators, but differ by the third logic operator, in which specific properties are actively used further. In spite of the fact that the Allen–Givone complete set is traditionally called Allen–Givone algebra (AGA), one should remember that this mathematical model differs greatly from the mass term algebra.

## 2.2. Definition of AGA and its Main Properties

The discrete k-valued AGA model was defined [71] as the complete set of non-Boolean operators

$$\langle 0, 1, \ldots, k-1, X(a,b), *, + \rangle \tag{1}$$

where the arbitrary function $y = f(x_1, \ldots, x_n)$ with $n$ input variables $x_1, \ldots, x_n$ and output variable $y$ can have $k$ discrete truth values, $x_1, x_2, \ldots, x_n, y \in L = \{0, 1, \ldots, k-1\}$. In expression (1):

- $0, 1, \ldots, k-1$ are the logical constants, i.e., $C_1, \ldots, C_{k-1} \in L = \{0, 1, \ldots, k-1\}$
- binary operator $\text{Min}(x_i, x_j)$ marked (*) acts on a pair of logic variables $x_i$ and $x_j$, choosing the minimal one,
- binary operator $\text{Max}(x_i, x_j)$ marked (+) acts on a pair of logic variables $x_i$ and $x_j$, choosing the maximal one,
- unary operator $X(a,b)$ is called Literal and acts on one logic variable $x$, where the result is given by expression (2):

$$X(a,b) = \begin{cases} 0, & if\ b < x < a \\ k-1, & if\ a \leq x \leq b \end{cases} \tag{2}$$

where for any $X(a,b)$ always $b \geq a$, and $a, b \in L = \{0, 1, \ldots, k-1\}$.

One can see from the Figure 3 and expression (2) that the operator Literal can be compared with a bandpass filter, where the lower and upper cutoff parameters can be adjusted. In addition, it can be realized as two sequential comparisons with thresholds $a$ and $b$.

The arbitrary MVL function $y = f(x_1, \ldots, x_n)$ always can be given by the truth table given in Figure 4, which has the overall number of rows $k^n - 1$, as the Boolean one can have only $2^n - 1$ rows. The column for output variable $y \in \{1, \ldots, k-1\}$ should be filled by some set of logical constants $C = \{0, 1, \ldots, k-1\}$. Every row of the truth table with a nonzero value of $y$ has equivalent representation, written as a product term, i.e., as logic expression, where logic constants and Literals are composed only via operators Min. Operator Literal (2) should be calculated for every input variable $x_j$ in every product term, resulting in $X_j(a, b)$ with some fixed values $a, b$. Thus, any product term for input variables $x_1, \ldots, x_n$ always has the structure $f(\ldots) * X_1(\ldots, \ldots) * X_2(\ldots, \ldots) * \ldots * X_n(\ldots, \ldots)$. Several expressions with this structure are given in the right side of Figure 3. According to the definitions in [71], the Literal $X(a, b)$ written for j-th input variable $x_j$ in the correspondent row of the truth table should be filled by equal values of parameters $a = b = \chi$, i.e., $X_j(\chi, \chi)$, where $\chi$ is the value of $x_j$ in the relevant row. (In row 1, e.g., $\chi$ has the value equal to 1).

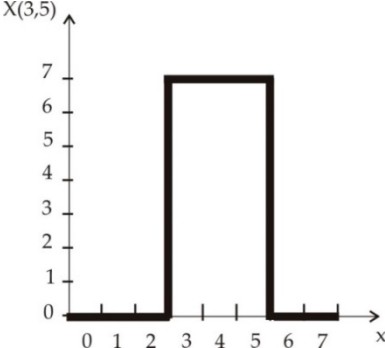

**Figure 3.** Logic operator Literal $X(a, b)$, shown for Allen–Givone algebra (AGA) with $k = 8$ truth levels and parameters $a = 3, b = 5$.

| $N_{row}$ | $x_1$ | $x_2$ | ... | $x_{n-1}$ | $x_n$ | $F(x_1,...,x_n)$ | corresponding product term |
|---|---|---|---|---|---|---|---|
| 0 | 0 | 0 | ... | 0 | 0 | $F(0,0,...,0)$ | $\longrightarrow F(0,0,...,0)*X_1(0,0)*X_2(0,0)*...*X_n(0,0)$ |
| 1 | 1 | 0 | ... | 0 | 0 | $F(1,0,...,0)$ | $\longrightarrow F(1,0,...,0)*X_1(1,1)*X_2(0,0)*...*X_n(0,0)$ |
| 2 | 2 | 0 | ... | 0 | 0 | $F(2,0,...,0)$ | $\longrightarrow F(2,0,...,0)*X_1(2,2)*X_2(0,0)*...*X_n(0,0)$ |
| ... | ... | ... | ... | ... | ... | ... | ... |
| $k^n$-1 | k-1 | k-1 | ... | k-1 | k-1 | $F(k-1,...,k-1)$ | $\longrightarrow F(k-1,k-1,...,k-1)*X_1(k-1,k-1)*X_2(k-1,k-1)*...*X_n(k-1,k-1)$ |

**Figure 4.** The truth table defines an arbitrary MVL function $y = f(x_1, \ldots, x_n)$. To the right of the table the set of product terms is shown, corresponding to appropriate rows.

Due to [71], the overall logic expression of an arbitrary function can be written as an expression where all nonzero product terms written right to the truth table are connected by operators Max (+). The resulting logical expression for $y = f(x_1, \ldots, x_n))$ will be

$$\begin{aligned} y = f(0, 0, \ldots, 0) * X_1(0, 0) * X_2(0, 0) * \ldots * X_n(0, 0) + \\ + f(0, 0, \ldots, 1) * X_1(1, 1) * X_2(0, 0) * \ldots * X_n(0, 0) + \ldots \\ + f(k-1, k-1, \ldots, k-1) * X_1(k-1, k-1) * X_2(k-1, k-1) * \ldots * X_n(k-1, k-1). \end{aligned} \tag{3}$$

The calculation of expression (3) and its product terms [71] should be done as follows. For every product term $Const * X_1(a_{m,1}, b_{m,1}) * \ldots * X_n(a_{m,n}, b_{m,n})$, all operators Literal $X_j(\ldots)$ should be calculated for the fixed set of values of input variables $\underline{x}_1, \ldots, \underline{x}_n$. If the current value of variable $x_j$ does not fit to the segment $[a_{m,j}, b_{m,j}]$, then $X_j(a_{m,j}, b_{m,j}) = 0$ and the whole product term will also be equal to 0.

Otherwise, $X_j\left(a_{m,j},\ b_{m,j}\right) = k-1$, and the product term will give the value $Min(k-1, Const) = Const$. Operators Max (+) will further give the greatest value from all the product terms in (3).

Logic expression (3) also provides the equivalent reverse transformation into its truth table [71].

The given above method [71] principally ensures the correct mathematical method to receive the formal expression for an arbitrary function with any finite number of rows. As it was discussed earlier [67,76], the structure of product terms easily provides parallel schemes for computing of (3). As the basic operators are the operations of comparison Max and Min, then the result of (3) will never overload the memory for any amount of data

The example of the program for the calculation of an arbitrary product term included in expression (3) with given parameters $k$ and $n$, is given in supplementary materials. This program was written in C language for debugging and tests of microassembler programs, emulating MVL models for microcontrollers MCS-51.

The application of large-scale MVL models for real IoT devices or cryptography tasks will need to represent the arbitrary MVL function as the set of matrixes [75]:

$$A_u = \begin{pmatrix} a_{11} & \dots & a_{1n} \\ \dots & \dots & \dots \\ a_{k-1,1} & \dots & a_{k-1,n} \end{pmatrix},\ B_u = \begin{pmatrix} b_{11} & \dots & b_{1n} \\ \dots & \dots & \dots \\ b_{k-1,1} & \dots & b_{k-1,n} \end{pmatrix},\ C = \begin{pmatrix} c_{11} & \dots & c_{1v} \\ \dots & \dots & \dots \\ c_{k-1,1} & \dots & c_{k-1,v} \end{pmatrix}, \tag{4}$$

where always $b_{ij} \geq a_{ij}$, $q$. In expression (4), all used matrix elements are natural numbers, where $u \in \{1, 2, \dots, v\}$ and $v$ is the number of blocks (or groups) of product terms, which can be fragmented during minimization. Matrixes $A_u$ and $B_u$ in expression (4) define indexed sets of Literal's parameters $a$ and $b$, respectively, and both of them have the fixed dimension $n \times (k-1)$. Namely, the number of non-zero elements in matrixes (4) should be transformed during the minimization procedure, as described in Section 2.3, and it is shown further that the information capacity of the MVL model is very high.

## 2.3. Multiparametrical Space Dimension for MVL Function

The task to integrate robotic systems into CN and to realize targeted language for their peer-to-peer communications requires designing the formal model of a high-dimensional space for the logical modeling of MAS systems. An arbitrary MVL function $y = f(x_1, \dots, x_n)$ given for n input variables and k truth levels can have up to $k^n$ rows in its truth table instead of $2^n$ in the Boolean logic [71,72], see Table 3. The number of all possible logic functions is much more: $k^{k^n}$ instead of $2^{2^n}$ for Boolean systems. Thus, the application of 8, 16, or 32 bit platforms for the emulation of MVL functions with several dozens of variables can provide an extremely high space of targeted parameters for MAS languages, which can principally include all the addressing spaces of CN and is represented as a sequence of bytes. Table 3 demonstrates the number of rows and logic functions for, e.g., functions of 50 input variables and for several values of $k$. In any way, even for only 16 bit platform, these dimensions exceed the number $4.63 \times 10^{170}$ of possible positions in Go play $19 \times 19$ [73], which is much less than the factor of expansion $e^{e^{10^{13}}} \approx 10^{1.55 \times 10^{4342944819032}}$ in the model of the expanding universe obtained in 1984 by Padmanabhan [74]. Some of the other largest known numbers were cited also in [80].

That is why MVL functions are principally much more effective than Boolean ones for the description of high-dimensional spaces, which are designed for targeted robotic communications. A further step is to propose the methods that are needed for the efficient computing of an MVL function in an agent or an MAS. The first step is to form the necessary MVL function and its logic expression. The simplest variant was disclosed in Section 2.1, where the number of rows in the truth table potentially can be extremely high. For such situations, one should be ready to simplify MVL function expression, thus shortening the computing time. This procedure resembles to some extent the minimization in Boolean logic, but it is much more cumbersome.

**Table 3.** Comparative information capacity of multiple-valued logic (MVL) truth table.

| N | K | $N_{rows} = k^n$ | $N_{log.functions} = k^{k^n}$ |
|---|---|---|---|
| 50 | 2 | $\approx 1.13 \times 10^{15}$ | $\approx 1.27 \cdot \times 10^{30}$ |
| 50 | 256 | $\approx 2.58 \cdot 10^{120}$ | $\approx 2.96 \cdot \times 10^{30825}$ |
| 50 | 65536 | $\approx 6.67 \cdot \times 10^{240}$ | $> 10^{15770928}$ |

*2.4. The Minimization Method for AGA Functions*

The simplification of expression (3) for an arbitrary function in AGA is based on the consensus method [71], which is the multi-stage sequential transformation of the whole set of parameters $a_{m,j}$ and $b_{m,j}$ in (3), decreasing the number of product terms. Useful additional commentaries are also given in Chapters 7–9 of the same book [71].

The consensus method to obtain the minimized logic expression for an MVL function is grounded on definitions 1–4, cited from [71] and disclosing the principle of subsuming product terms and the application of the consensus method.

**Definition 1.** *Product term $r_1 * X_1(a_1, b_1) * \ldots * X_n(a_n, b_n)$ subsumes another product term $r_2 * X_1(c_1, d_1) * \ldots * X_n(c_n, d_n)$, if and only if both conditions are true:*

$$1)\ r_1 \leq r_2 \tag{5}$$

$$2)\ c_i \leq a_i \leq b_i \leq d_i \text{ for all } X_i,\ i = 1, \ldots, n.$$

**Example 1.** *Let $k = 60$ and variables $x_1, x_2, x_3$ are defined. The product term $1 * X_1(1,1) * X_2(12,25) * X_3(37,44)$ subsumes term $2 * X_1(0,7) * X_2(10,41) * X_3(35,59)$, as its parameters $a_i, b_i$ are between $c_i$ and $d_i$, and*

*for $x_1$, inequalities $0 < a = 1$ and $b = 1 < 7$ are true,*
*for $x_2$, we have $10 < a = 12$, $b = 25 < 41$ and*
*for $x_3$, expressions are true $35 < a = 37$, $b = 44 < 59$.*
*As in the notation* ~~$1 * X_1(1,1) * X_2(12,25) * X_3(37,44)$~~ $+ 2 * X_1(0,7) * X_2(10,41) * X_3(35,59)$, *the first product term is included into the second one and can be deleted, thus it was crossed out.*

The idea of the minimization procedure is based on the search for subsuming product terms, shortening the logic expression. If all the subsumed product terms were found, then the consensus operation [71] is to be applied.

**Definition 2.** ***Consensus*** *$j * U_j *^i k \cdot U_k$ in the $i$–th coordinate for product terms*
*$r * U_1 = r * X_1(a_1, b_1) * \ldots * X_n(a_n, b_n)$ and $s * U_2 = s * X_1(c_1, d_1) * \ldots * X_n(c_n, d_n)$ is given by the expression*
*$j * U_j *^i k * U_k = q * X_1(e_1, f_1) * X_2(e_2, f_2) \ldots * X_n(e_n, f_n)$* ***if and only if*** *there exists such a set of $q, e_k, f_k$ that*

$$q = j * k \tag{6}$$

$X_k(e_k, f_k) = X_k(a_k, b_k) + X_k(c_k, d_k)$ *for $k = i$,*
$X_k(e_k, f_k) = X_k(a_k, b_k) * X_k(c_k, d_k)$ *for all $k \neq i$.*

In expression (5), two additional operations are used, which are disclosed further.

**Definition 3.** *The operator **union** of Literals is defined as* $X(a,b) = X(c,d) + X(e,f)$, *and it exists if*

$$a = MIN(c,e), \quad b = MAX(d,f), \quad e - 1 \le d, \quad c - 1 \le f \tag{7}$$

**Definition 4.** *The **intersection** of Literals is defined as* $X(a,b) = X(c,d) * X(e,f)$, *and it exists if*

$$a = MAX(c,e), \quad b = MIN(d,f), \quad e \le d, \quad c \le f. \tag{8}$$

These operations are shown in Figure 5 and respond to the calculation of expressions $X_1(1,4) * X_2(2,6) = X(2,4)$ and $X_1(1,4) + X_2(2,6) = X(1,6)$. Simplified visual interpretation for the intersection and union of Literals can be proposed, basing on differently featured rectangles (marked gray), which fill the band between parameters $a$ and $b$ in Literals $X(a,b)$. For two product terms, the consensus in the i-th input variable can be calculated if there is union, i.e., a common (not fragmented) rectangular for both Literals $X_i(a,b)$, marked by gray in the case b in Figure 5b, and also there exists an intersection of rectangles for both Literals, which is marked by gray in case a, and it is taken for all the other variables.

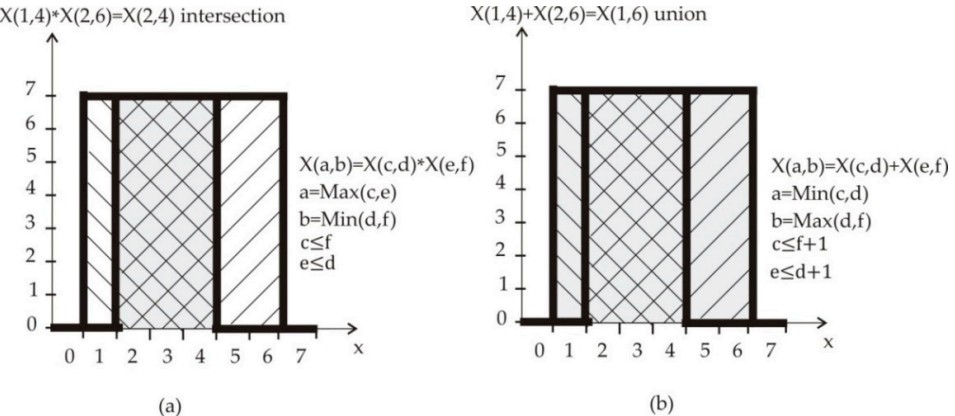

**Figure 5.** Operation of "intersection" and "union" used in the minimization procedure. (**a**): Intersection of two Literals is given by the coincident set of their values x, marked by grey. (**b**) Union of two Literals is given by the enlarged common set of their values x, marked by grey.

Example 2 further demonstrates the calculation of consensus for two product terms.

**Example 2.** *Lets us calculate the consensus for product terms* $2 * X_1(1,3) * X_2(0,1)$ *and* $3 * X_1(3,4) * X_2(2,3) * X_3(2,4)$, *which is given for three input variables and five truth levels.*

As the variable $x_3$ is not used in the first product term, according to [71], it should be rewritten as $2 * X_1(1,3) * X_2(0,1) * X_3(0,4)$, where Literal $X_3(0, k-1) = X_3(0,4)$ is added.

Consensus $2 * X_1(1,3) * X_2(0,1) * X_3(0,4) *^1 3 * X_1(3,4) * X_2(2,3) * X_3(2,4)$, taken in $x_1$, does not exist as there are no such $e$ and $f$, for which $X_2(e,f) = X_2(0,1) * X_2(2,3)$ and rectangle markers do not intersect.
Consensus $2 * X_1(1,3) * X_2(0,1) * X_3(0,4) *^2 3 * X_1(3,4) * X_2(2,3) * X_3(2,4)$ taken in $x_2$ exists, as there are such $e$ and $f$ that $X_2(e,f) = X_2(0,1) + X_2(2,3)$.

The procedure for the minimization of given MVL functions was defined in [71], and it was further commented on in [107,108]. It is based on the use of so-called "don't care states" (DCS), which are the non-defined or vacuous rows of the truth table, where $y$ column cells should be filled by a $k-1$ constant. If the column $y$ in the row is intentionally filled by zero, this row should not be taken as a

DCS, as this will change the initially given function. Thus, the substitution of a DCS into an initially given MVL function redefines the extended variant of the MVL function, which reproduces all the output values of the initially given one. From the point of view of mathematics, such a minimization method for logic function does not create any problems. However, for applied tasks, where logic variables respond to real physical parameters of hardware actuators, the reckless choice of DCS = $k - 1$ can influence the work of sloppy designed subsystems. Thus, in a general case, the choice of DCS requires special attention and is explained further for the classifier module.

MVL function minimization can be represented as the algorithm described in detail in [71].

- Algorithm of minimization of AGA function.

Preliminary given parameters: $k$-number of truth levels, n-number of input variables.

1.  Form a numbered list of product terms for the given truth table (or the logic expression). (Note. Principally, the order does not influence the final result, but not completing the minimization procedure enlarges the computing time).
2.  Replace all vacuous Literals by Literals $X_i(0, k-1)$.
3.  Remove all subsuming product terms in the list, using expression (4) for all possible pairs of product terms.
4.  Add one or several DCS, substituting $k - 1$ instead of undefined rows in the truth table, add newly formed product terms to the list, and assign consecutive numbers to product terms.
5.  Repeat the search for subsuming product terms, delete such terms from the list, and assign the numbers in the obtained list.
6.  Compute consensus by expression (5) sequentially for all input variables and product terms; if the obtained product terms do not subsume other terms, add them to the bottom of the list.
7.  Find all the prime implicants that do not subsume more other terms.
8.  Delete all literals of the form $X_i(0, k-1)$ in every product term (*since they are equal to $k - 1$ by definition*).

The given above method is bulky and cumbersome, even for small-scale examples [107,108].

Typical problems of the MVL minimization are demonstrated for the model function $F(x_1, x_2, x_3)$, which is defined in Table 4, where for simplicity, the minimization procedure is done for the simple fragment of some truth table, given for $k = 255$. In real robotic tasks, such function, e.g., can represent the results of monitoring atmosphere pollution, where the drone measures the pollution concentration during flight with the constant altitude, e.g., $x_3 = 3$ a.u. Here, $x_1, x_2$ refers to surface coordinates $x$ and $y$. N is the number of the term. The product terms in the right part of Table 4 respond to every row of the table.

**Table 4.** Truth table for the function F($x_1, x_2, x_3$).

| N | $X_1$ | $X_2$ | $X_3$ | F($x_1, x_2, x_3$) | Responding Product Terms |
|---|---|---|---|---|---|
| 1 | 2 | 0 | 3 | 75 | 75*$X_1$(2,2)* $X_2$(0,0)* $X_3$(3,3) |
| 2 | 1 | 1 | 3 | 11 | 11*$X_1$(1,1)*$X_2$(1,1)* $X_3$(3,3) |
| 3 | 2 | 1 | 3 | 75 | 75*$X_1$(2,2)* $X_2$(1,1)*$X_3$(3,3) |
| 4 | 1 | 2 | 3 | 11 | 11*$X_1$(1,1)*$X_2$(2,2)*$X_3$(3,3) |
| 5 | 2 | 2 | 3 | 75 | 75*$X_1$(2,2)*$X_2$(2,2)*$X_3$(3,3) |
| 6 | 3 | 2 | 3 | 11 | 11*$X_1$(3,3)*$X_2$(2,2)*$X_3$(3,3) |
| 7 | 2 | 3 | 3 | 75 | 75*$X_1$(2,2)*$X_2$(3,3)*$X_3$(3,3) |
| 8 | 3 | 3 | 3 | 5 | 5*$X_1$(3,3)*$X_2$(3,3)*$X_3$(3,3) |
| 9 | 3 | 4 | 3 | 5 | 5* $X_1$(3,3)*$X_2$(4,4)*$X_3$(3,3) |

The obtained set of product terms is rewritten for visual aids:

$$
\begin{array}{cl}
1 & 5^*X_1(3,3)^*X_2(3,3)^*X_3(3,3) \\
2 & 5^*X_1(3,3)^*X_2(4,4)^*X_3(3,3) \\
3 & 11^*\,X_1(1,1)^*X_2(1,1)^*X_3(3,3) \\
4 & 11^*X_1(1,1)^*X_2(2,2)^*X_3(3,3) \\
5 & 11^*X_1(3,3)^*X_2(2,2)^*X_3(3,3) \\
6 & 75^*X_1(2,2)^*X_2(0,0)^*X_3(3,3) \\
7 & 75^*X_1(2,2)^*X_2(1,1)^*X_3(3,3) \\
8 & 75^*X_1(2,2)^*X_2(2,2)^*X_3(3,3) \\
9 & 75^*X_1(2,2)^*X_2(3,3)^*X_3(3,3).
\end{array}
$$

Using the definition in expression (5), one can see that there are no subsuming terms in the list given above.

According to the algorithm of minimization given above, Steps 1 and 2 are not activated here. Due to Step 3, the search for consensus is based on definition of expression. (5) and should begin as follows:

consensus $1*^1 2$ does not exist,

consensus $1*^2 2 = 5^*X_1(3,3)^*X_2(3,3)^*X_3(3,3) \,*^2\, 5^*X_1(3,3)^*X_2(4,4)^*X_3(3,3) = 5^*X_1(3,3)^*X_2(3,4)^*X_3(3,3)$, so that

| | $1*^2 2$ is added as term 10 | | terms 1 and 2 subsume 10 | | terms 1 and 2 are deleted |
|---|---|---|---|---|---|
| 1 | $5^*X_1(3,3)^*X_2(3,3)^*X_3(3,3)$ | 1 | ~~$5^*X_1(3,3)^*X_2(3,3)^*X_3(3,3)$~~ | 1 | $11^*X_1(1,1)^*X_2(1,1)^*\,X_3(3,3)$ |
| 2 | $5^*X_1(3,3)^*X_2(4,4)^*X_3(3,3)$ | 2 | ~~$5^*X_1(3,3)^*X_2(4,4)^*X_3(3,3)$~~ | 2 | $11^*X_1(1,1)^*X_2(2,2)^*X_3(3,3)$ |
| 3 | $11^*X_1(1,1)^*X_2(1,1)^*X_3(3,3)$ | 3 | $11^*X_1(1,1)^*X_2(1,1)^*X_3(3,3)$ | 3 | $11^*X_1(3,3)^*X_2(2,2)^*X_3(3,3)$ |
| 4 | $11^*X_1(1,1)^*X_2(2,2)^*X_3(3,3)$ | 4 | $11^*X_1(1,1)^*X_2(2,2)^*X_3(3,3)$ | 4 | $75^*X_1(2,2)^*X_2(0,0)^*X_3(3,3)$ |
| 5 | $11^*X_1(3,3)^*X_2(2,2)^*X_3(3,3)$ | 5 | $11^*X_1(3,3)^*X_2(2,2)^*X_3(3,3)$ | 5 | $75^*X_1(2,2)^*X_2(1,1)^*X_3(3,3)$ |
| 6 | $75^*X_1(2,2)^*X_2(0,0)^*X_3(3,3)$ | 6 | $75^*X_1(2,2)^*X_2(0,0)^*X_3(3,3)$ | 6 | $75^*X_1(2,2)^*X_2(2,2)^*X_3(3,3)$ |
| 7 | $75^*X_1(2,2)^*X_2(1,1)^*X_3(3,3)$ | 7 | $75^*X_1(2,2)^*X_2(1,1)^*X_3(3,3)$ | 7 | $75^*X_1(2,2)^*\,X_2(3,3)^*X_3(3,3)$ |
| 8 | $75^*X_1(2,2)^*X_2(2,2)^*X_3(3,3)$ | 8 | $75^*X_1(2,2)^*X_2(2,2)^*X_3(3,3)$ | 8 | $5^*X_1(3,3)^*\,X_2(3,4)^*X_3(3,3).$ |
| 9 | $75^*X_1(2,2)^*X_2(3,3)^*X_3(3,3)$ | 9 | $75^*X_1(2,2)^*X_2(3,3)^*X_3(3,3)$ | | |
| 10 | $5^*X_1(3,3)^*X_2(3,4)^*X_3(3,3)$ | 10 | $5^*X_1(3,3)^*\,X_2(3,4)^*X_3(3,3)$ | | |

A new search for the consensus provides:

consensus $1*^1 2$ does not exist,

consensus $1*^2 2 = 11^*\,X_1\,(1,1)^*x2(1,1)^*x3(3,3)\,*^1\,11^*\,X_1\,(1,1)\,*\,x2(2,2)\,x3(3,3) = 11^*\,X_1\,(1,1)^*X2(1,2)^*X3(3,3)$,

| | $1*^2 2$ is added as term 9 | | terms 1 and 2 subsume 9 | | terms 1 and 2 are deleted |
|---|---|---|---|---|---|
| 1 | $11^*X_1(1,1)\,^*X_2(1,1)^*X_3(3,3)$ | 1 | ~~$11^*X_1(1,1)^*X_2(1,1)^*\,X_3(3,3)$~~ | 1 | $11^*X_1(3,3)^*X_2(2,2)^*X_3(3,3)$ |
| 2 | $11^*X_1(1,1)^*X_2(2,2)^*X_3(3,3)$ | 2 | ~~$11^*X_1(1,1)^*X_2(2,2)^*X_3(3,3)$~~ | 2 | $75^*X_1(2,2)^*X_2(0,0)^*X_3(3,3)$ |
| 3 | $11^*X_1(3,3)^*X_2(2,2)^*X_3(3,3)$ | 3 | $11^*X_1(3,3)\,^*X_2(2,2)^*X_3(3,3)$ | 3 | $75^*X_1(2,2)^*X_2(1,1)^*X_3\,(3,3)$ |
| 4 | $75^*X_1(2,2)^*X_2(0,0)^*X_3(3,3)$ | 4 | $75^*X_1(2,2)^*X_2(0,0)^*X_3(3,3)$ | 4 | $75^*X_1(2,2)^*X_2(2,2)^*X_3(3,3)$ |
| 5 | $75^*X_1(2,2)^*X_2(1,1)^*X_3(3,3)$ | 5 | $75^*X_1(2,2)^*X_2(1,1)^*X_3(3,3)$ | 5 | $75^*X_1(2,2)^*X_2(3,3)^*X_3(3,3)$ |
| 6 | $75^*X_1(2,2)^*X_2(2,2)^*X_3(3,3)$ | 6 | $75^*X_1(2,2)^*X_2(2,2)^*X_3(3,3)$ | 6 | $5^*X_1(3,3)^*X_2(3,4)^*X_3(3,3)$ |
| 7 | $75^*X_1(2,2)^*X_2(3,3)^*\,X_2\,(3,3)$ | 7 | $75^*X_1(2,2)^*X_2(3,3)^*X_3(3,3)$ | 7 | $11^*X_1(1,1)^*X_2(1,2)^*X_3\,(3,3).$ |
| 8 | $5^*X_1(3,3)^*X_2(3,4)^*X_3(3,3)$ | 8 | $5^*X_1(3,3)^*X_2(3,4)^*X_3(3,3)$ | | |
| 9 | $11^*X_1(1,1)^*X_2(1,2)^*\,X_3(3,3)$ | 9 | $11^*X_1(1,1)^*X_2(1,2)^*X_3\,(3,3)$ | | |

A further procedure demonstrates that term 1 has no consensuses with terms 2–7, so one should check for term 2:

consensus $2*^1 3$ does not exist,

consensus $2*^2 3 = 2*^2 3 = 75^*x1(2,2)^*x2(0,0)\,*\,x3(3,3)\,*^2\,75^*x1(2,2)^*x2(1,1)\,*\,x3(3,3) = 75^*X1(2,2)^*X2(0,1)^*X3(3,3)$,

2*²3 is added as term 8 | | terms 2 and 3 subsume 8 | | terms 1 and 2 are deleted

| | 2*²3 is added as term 8 | | terms 2 and 3 subsume 8 | | terms 1 and 2 are deleted |
|---|---|---|---|---|---|
| 1 | $11*X_1(3,3)*X_2(2,2)*X_3(3,3)$ | 1 | $11*X_1(3,3)*X_2(2,2)*X_3(3,3)$ | 1 | $11*X_1(3,3)*X_2(2,2)*X_3(3,3)$ |
| 2 | $75*X_1(2,2)*X_2(0,0)*X_3(3,3)$ | 2 | ~~$75*X_1(2,2)*X_2(0,0)*X_3(3,3)$~~ | 2 | $75*X_1(2,2)*X_2(2,2)*X_3(3,3)$ |
| 3 | $75*X_1(2,2)*X_2(1,1)*X_3(3,3)$ | 3 | ~~$75*X_1(2,2)*X_2(1,1)*X_3(3,3)$~~ | 3 | $75*X_1(2,2)*X_2(3,3)*X_3(3,3)$ |
| 4 | $75*X_1(2,2)*X_2(2,2)*X_3(3,3)$ | 4 | $75*X_1(2,2)*X_2(2,2)*X_3(3,3)$ | 4 | $5*X_1(3,3)*X_2(3,4)*X_3(3,3)$ |
| 5 | $75*X_1(2,2)*X_2(3,3)*X_3(3,3)$ | 5 | $75*X_1(2,2)*X_2(3,3)*X_3(3,3)$ | 5 | $11*X_1(1,1)*X_2(1,2)*X_3(3,3)$ |
| 6 | $5*X_1(3,3)*X_2(3,4)*X_3(3,3)$ | 6 | $5*X_1(3,3)*X_2(3,4)*X_3(3,3)$ | 6 | $75*X_1(2,2)*X_2(0,1)*X_3(3,3),$ |
| 7 | $11*X_1(1,1)*X_2(1,2)*X_3(3,3)$ | 7 | $11*X_1(1,1)*X_2(1,2)*X_3(3,3)$ | | |
| 8 | $75*X_1(2,2)*X_2(0,1)*X_3(3,3)$ | 8 | $75*X_1(2,2)*X_2(0,1)*X_3(3,3)$ | | |

Consensus $2*²3 = 75*x1(2,2)*X_2(2,2)*x3(3,3)*²75*x1(2,2)*X_2(3,3)*x3(3,3) = 75*X1(2,2)*X_2(2,3)*X3(3,3),$

| | 2*²3 is added as term 7 | | terms 2 and 3 subsume 7 | | terms 1 and 2 are deleted |
|---|---|---|---|---|---|
| 1 | $11*X_1(3,3)*X_2(2,2)*X_3(3,3)$ | 1 | $11*X_1(3,3)*X_2(2,2)*X_3(3,3)$ | 1 | $11*X_1(3,3)*X_2(2,2)*X_3(3,3)$ |
| 2 | $75*X_1(2,2)*X_2(2,2)*X_3(3,3)$ | 2 | ~~$75*X_1(2,2)*X_2(2,2)*X_3(3,3)$~~ | 2 | $5*X_1(3,3)*X_2(3,4)*X_3(3,3)$ |
| 3 | $75*X_1(2,2)*X_2(3,3)*X_3(3,3)$ | 3 | ~~$75*X_1(2,2)*X_2(3,3)*X_3(3,3)$~~ | 3 | $11*X_1(1,1)*X_2(1,2)*X_3(3,3)$ |
| 4 | $5*X_1(3,3)*X_2(3,4)*X_3(3,3)$ | 4 | $5*X_1(3,3)*X_2(3,4)*X_3(3,3)$ | 4 | $75*X_1(2,2)*X_2(0,1)*X_3(3,3)$ |
| 5 | $11*X_1(1,1)*X_2(1,2)*X_3(3,3)$ | 5 | $11*X_1(1,1)*X_2(1,2)*X3(3,3)$ | 5 | $75*X_1(2,2)*X_2(2,3)*X_3(3,3).$ |
| 6 | $75*X_1(2,2)*X_2(0,1)*X_3(3,3)$ | 6 | $75*X_1(2,2)*X_2(0,1)*X_3(3,3)$ | | |
| 7 | $75*X_1(2,2)*X_2(2,3)*X_3(3,3)$ | 7 | $75*X_1(2,2)*X_2(2,3)*X_3(3,3)$ | | |

Consensus $4*²5 = 75*x1(2,2)*x2(0,1)*x3(3,3)*2\ 75*x1(2,2)*x2(2,3)*x3(3,3) = 75*X1(2,2)*X2(0,3)*X3(3,3)$

| | 4*²5 is added as term 6 | | terms 4 and 5 subsume 6 | | terms 4 and 5 are deleted |
|---|---|---|---|---|---|
| 1 | $11*X_1(3,3)*X_2(2,2)*X_3(3,3)$ | 1 | $11*X_1(3,3)*X_2(2,2)*X_3(3,3)$ | 1 | $11*X_1(3,3)*X_2(2,2)*X_3(3,3)$ |
| 2 | $5*X_1(3,3)*X_2(3,4)*X_3(3,3)$ | 2 | $5*X_1(3,3)*X_2(3,4)*X_3(3,3)$ | 2 | $5*X_1(3,3)*X_2(3,4)*X_3(3,3)$ |
| 3 | $11*X_1(1,1)*X_2(1,2)*X_3(3,3)$ | 3 | $11*X_1(1,1)*X_2(1,2)*X_3(3,3)$ | 3 | $11*X_1(1,1)*X_2(1,2)*X_3(3,3)$ |
| 4 | $75*X_1(2,2)*X_2(0,1)*X_3(3,3)$ | 4 | ~~$75*X_1(2,2)*X_2(0,1)*X_3(3,3)$~~ | 4 | $75*X_1(2,2)*X_2(0,3)*X_3(3,3)$ |
| 5 | $75*X_1(2,2)*X_2(2,3)*X_3(3,3)$ | 5 | ~~$75*X_1(2,2)*X_2(2,3)*X_3(3,3)$~~ | | |
| 6 | $75*X_1(2,2)*X_2(0,3)*X_3(3,3)$ | 6 | $75*X_1(2,2)*X_2(0,3)*X_3(3,3)$ | | |

Thus, the overall number of terms was shortened by the minimization procedure from 9 to 4.

Principally, the final set of terms should represent the list of prime implicants, which can not be shortened further. Formally, the minimization procedure is not finished until all the possible DCS are not substituted. The obtained intermediate set of four product terms contains two terms with constants $C = 11$, which seems to be potentially transformed further for appropriately chosen DCS with $C = 255$; this is substituted in place of the undefined rows in the truth table. This choice of DCS is not reglamented in [71]. However, for the function with $k = 256$ and dozens of input variables, the number of possible DCS is too large. It is physically impossible to substitute DCS into all unused rows! The only reasonable way here is to design further the special AI algorithm for the intentional choice of DCS.

The general motivation of such an intentional search is shown in detail in Figure 6 for the results of simplified 2D mapping of the MVL function, which is defined in Table 4 and minimized above. The interliminary list of obtained product terms includes four terms. For final minimization, the choice of DCS coordinates $(x_1, x_2, x_3) = (3, 1, 3)$ (case c) in Figure 6 is more preferable than $(x_1, x_2, x_3) = (1, 3, 3)$ (case b) in Figure 6), although both of them symmetrically fill corner points for prime implicants for mapping $(x_1, x_2)$.

For the case DCS = (3,1,3), one should take the interliminary set of four product terms and add the term $255*X_1(3,3)*X_2(1,1)*X_3(3,3)$ to the end of list:

| 1 | $11*X_1(3,3)*X_2(2,2)*X_3(3,3)$ |
|---|---|
| 2 | $5*X_1(3,3)*X_2(3,4)*X_3(3,3)$ |
| 3 | $11*X_1(1,1)*X_2(1,2)*X_3(3,3)$ |
| 4 | $75*X_1(2,2)*X_2(0,3)*X_3(3,3)$ |
| 5 | $255*X_1(3,3)*X_2(1,1)*X_3(3,3)$ |

A new search for consensus provides

Consensus $1*^25 = 11 * X_1 (3,3) * X_2 (2,2) * X_3 (3,3) *^2 255* X_1 (3,3) * X_2 (1,1) * X_3 (3,3) = 255 * X_1 (3,3) * X_2 (0,3) * X_3 (3,3)$.

| | $1*^25$ is added as term 6 | | terms 1 and 5 subsume 6 | | terms 1 and 5 are deleted |
|---|---|---|---|---|---|
| 1 | $11*X_1(3,3)*X_2(2,2)*X_3(3,3)$ | 1 | ~~$11*X_1(3,3)*X_2(2,2)*X_3(3,3)$~~ | 1 | $5*X_1(3,3)*X_2(3,4)*X_3(3,3)$ |
| 2 | $5*X_1(3,3)*X_2(3,4)*X_3(3,3)$ | 2 | $5*X_1(3,3)*X_2(3,4)*X_3(3,3)$ | 2 | $11*X_1(1,1)*X_2(1,2)*X_3(3,3)$ |
| 3 | $11*X_1(1,1)*X_2(1,2)*X_3(3,3)$ | 3 | $11*X_1(1,1)*X_2(1,2)*X_3(3,3)$ | 3 | $75*X_1(2,2)*X_2(0,3)*X_3(3,3)$ |
| 4 | $75*X_1(2,2)*X_2(0,3)*X_3(3,3)$ | 4 | $75*X_1(2,2)*X_2(0,3)*X_3(3,3)$ | 4 | $255*X_1(3,3)*X_2(0,3)*X_3(3,3)$ |
| 5 | $255*X_1(3,3)*X_2(1,1)*X_3(3,3)$ | 5 | ~~$255*X_1(3,3)*X_2(1,1)*X_3(3,3)$~~ | | |
| 6 | $255*X_1(3,3)*X_2(0,3)*X_3 (3,3)$ | 6 | $255*X_1(3,3)*X_2(0,3)*X_3(3,3)$ | | |

That result of the last calculation does not shorten the number of prime implicants (still equal to 4), but formally, this variant, as shown in Figure 6c, is more correct, as the large ellipse describes all cells with $C = 11$ as one of the largest rectangular clusters of cells, containing only $C = 11$ or greater constants. However, the choice of DCS = (1,3,3) and the addition of the term $255* X_1(3,3) * X_2(1,1) * X_3(3,3)$ to the intermediary list of product terms given in Figure 6a provides

| | |
|---|---|
| 1 | $11*X_1(3,3)*X_2(2,2)*X_3(3,3)$ |
| 2 | $5*X_1(3,3)*X_2(3,4)*X_3(3,3)$ |
| 3 | $11*X_1(1,1)*X_2(1,2)*X_3(3,3)$ |
| 4 | $75*X_1(2,2)*X_2(0,3)*X_3(3,3)$ |
| 5 | $255*X_1(3,3)*X_2(1,1)*X_3(3,3)$ |

which does not give new consensus.

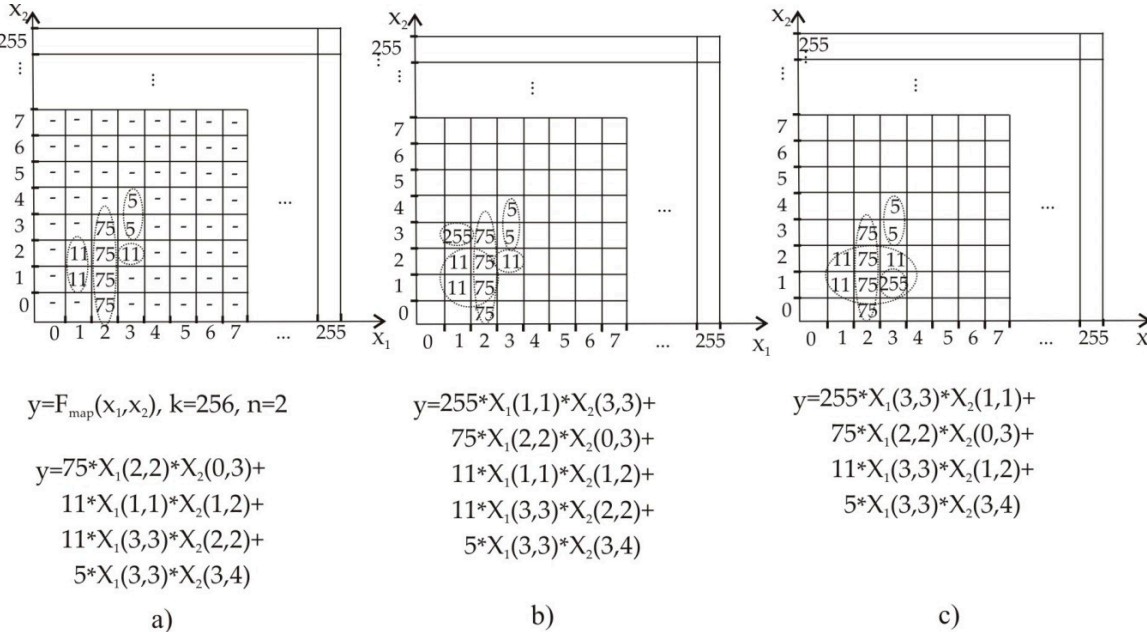

**Figure 6.** The impact of the choice of "don't care states" (DCS) on the final result of minimization of the MVL function given in Table 4. Undefined states DCS are marked "–". (**a**) The intermediary set of obtained product terms is shown for simplicity as 2D mapping $F_{map}(x_1, x_2)$, and it is given only for input variables $x_1, x_2$, as $x_3$ is constant. Obtained product terms are marked by ellipses and by product terms. (**b**) The 2D set of prime implicants, which was obtained by the minimization for the DCS = (1,3,3). Such a choice enlarges the list of product terms by one term $255* X_1(1,1) * X_2(3,3)$. (**c**) The choice of DCS = $(3, 1, 3)$ correctly finishes the search of prime implicants.

From the practical point of view, it is possible to use both variants with four prime implicants. However, the more preferable variant is to foresee by additional AI algorithms the features of the

situation, when the full minimization is of no sense. Principally, it is possible to count the events with the enlargement of product terms and then to return back to the earlier obtained version with four prime implicants, thus excluding the product term 255* $X_1(3,3)$ * $X_2(1,1)$ * $X_3(3,3)$ from the list. However, this procedure can not be described only within MVL models and should use external AI procedures, which is one of reasons to discuss special heterogeneous architecture in Section 3.

Thus, the formal consensus minimization of the MVL function in Table 4 is not the trivial procedure and can lead to different final results depending on the choice of DCS and the real number of runs carried out for the minimization algorithm. Of course, the random choice of the limited number of DCS does not guarantee the best result and needs to be analyzed more thoroughly. As it was shown in [108], the choice of DCS should be regarded as a multi-criteria task, which for real physical variables should be done elaborately, basing on Pareto optimization methods. Potentially, methods of image processing seem to be useful here, too.

The above examples demonstrate that the analysis of product terms and prime implicants provides the principal possibility of using logic minimization as the tool for analyzing the structure of logic constants for arbitrarily given logic switching functions. Although MVL minimization [71] itself needs external intellectual algorithms, its basic advantage is the correct method of equivalent transformations of any logic expression without any information losses. DCS-based minimization can be applied for correct transformations and the analysis of arbitrary types of aggregated data, which makes it a potential candidate for the processing and clusterization of mixed types of data, including both numerical and categorical data [90].

It is necessary to emphasize that one of the reasons to use further special architecture is determined by the need to analyze more complicated cases than the MVL function in Table 4. If e.g., the drone measures atmospheric pollutions at different altitudes $x_3$, then the structure of prime implicants would be much more complicated, which will need a special algorithm to process and analyze it.

## 3. Classification Schemes Based on MVL Functions

### 3.1. AGA Classifier and its Learning With a Teacher

The classification of objects [83] is regarded as the task to design the algorithm (or the device) that is capable of finding the arbitrary object $x \in X$ (taken from the set $X$ of input instances) and the corresponding tag (name) of its class $y \in Y$, where $Y$ is the set of output labels. The classification should be based on known learning samples $X^m = \{ (x_1, y_1), \ldots, (x_m, y_m) \}$.

The MVL function can be easily learned with a teacher to recognize $q$ objects. Let's take $q$ objects $O_q$, which are known to belong to $J$ different classes $\ell_j$. The set of classification parameters $x_1, \ldots, x_n$ is supposed to be enough for the classification of all objects. Thus, objects respond to $q$ rows of the truth table in Figure 3, for which the logic constant in $y$ column is equal to some nonzero value $\ell_j$. Then, the classification algorithm (or a final device) is defined by some function

$$
\begin{aligned}
y = F_c(x_1, \ldots, x_n) = \quad & C_1 * X_1(a_{11}, b_{11}) * X_2(a_{12}, b_{12}) * \ldots * X_n(a_{1n}, b_{1n}) + \\
& C_2 * X_1(a_{21}, b_{21}) * X_2(a_{22}, b_{22}) * \ldots * X_n(a_{2n}, b_{2n}) + \\
& \cdots \\
& C_{k-1} * X_1(a_{k-1,1}, b_{k-1,1}) * X_2(a_{k-1,2}, b_{k-1,2}) * \ldots * X_n(a_{k-1,n}, b_{k-1,n}),
\end{aligned}
\tag{9}
$$

which reproduces the output logic value (or the class tag) $\ell_j$ for every combination of input variables $x_1, \ldots, x_n$ taken from the given set.

As the MVL function [75] can be defined with an arbitrary number of truth levels $k$, then the switching logical function $y = F_c(x_1, \ldots, x_n)$ always can be generated to reproduce the mentioned above classifier. This can be done correctly for any finite number of objects and classes, as the MVL truth table by its definition [75] provides the mathematically correct method to receive the logic expression for an arbitrary function, which is written as a finite number of rows in the truth table.

### 3.2. Nested Structure of Classes in AGA Classifier

The classifier based on AGA functions will provide the nested structure of classes $\mathcal{C}_j$, as its minimization procedure always produce prime implicants with appropriate logic constants according to the scale of truth levels $C_j$. This nested structure is shown in Figure 7. That is why the structure of prime implicants is hierarchical and ordered, i.e., the object with a greater $C_j$ value will be always included into the n-dimensional rectangular "box" with a smaller $C_j$ value as an object with a greater true level. One should keep in mind this specific feature while designing the classification tree. For example, the objects can be incomparable within the context of the robotic task, but their tags of classes can respond to the material cost of objects. The general structure of classes here can differ from the scale of truth values. The AGA can be defined for $k = 256$ and greater, but in reality, one may use only one part of tags, reserving the structure for complicated schemes with network mapping of MVL functions [76].

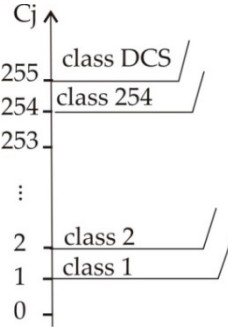

**Figure 7.** AGA classification by definition has the nested structure of classes $\mathcal{C}_j$, responding to truth levels scale $C_j$.

As it follows from [71], the maximal logic constant $C_j = k - 1$ always should be substituted instead of DCS for the minimization proceedure. In a model with the large numbers of variables $n$ and truth levels $k - 1$, it is reasonable to reserve the maximal $C_j$ only for the tagging of DCS. It will help to monitor the steps of the minimization procedure. In order to learn such a function (with a teacher), one should compose the table as in Figure 3 according to algorithm as follows.

- Algorithm of Learning MVL Classifier with a Teacher

1.  To evaluate the maximal number of objects (tags) $q$, i.e., to fix the necessary number of rows in the truth table.
2.  To evaluate the maximal number of classes $J$ to be used, thus choosing the necessary number of truth levels $k > J$.
3.  To write the list of all characteristic features for classification $n$, thus choosing the number of input variables $x_1, \ldots, x_n$.
4.  To fill the rows of the table by sets $x_1, \ldots, x_n$.
5.  To fill the column for $y = F(x_1, \ldots, x_n)$ by responding tags of classes $C_i$, using any correct methods for their estimates.
6.  To compose the expression набор $C_j * X_1(x_{i,1}, b_{i,1}) * \ldots * X_n(a_{i,n}, b_{i,n})$ for every row $x_1, \ldots, x_n$.
7.  To unite all sets of minterms $C_j * X_1(x_{i,1}, b_{i,1}) * \ldots * X_n(a_{i,n}, b_{i,n})$ by the operator MAXIMUM (+) into the general expression of the function

$$
\begin{aligned}
y = & f(0,0,\ldots,0) * X_1(0,0) * X_2(0,0) * \ldots * X_n(0,0) + \\
& + f(0,0,\ldots,1) * X_1(1,1) * X_2(0,0) * \ldots * X_n(0,0) + \\
& \ldots \\
& + f(k-1,k-1,\ldots,k-1) * X_1(k-1,k-1) * X_{j2}(k-1,k-1) * \ldots * X_{jn}(k-1,k-1).
\end{aligned}
\tag{10}
$$

8. To hold a consensus minimization procedure according to Section 2, in order to reduce the computing time.
9. To choose the platform and adapt the software for emulation of the minimized MVL function by the classifier device.

The AGA method of formal description of hierarchical structures potentially provides the possibility to process the unknown set of data, basing only on the chosen set of classification features, but it does not need to use any hypothesis concerning the density of objects or statistic parameters. That is the principal potential advantage of MVL classifier schemes and is the task for future investigations.

### 3.3. Description of Objects in the AGA Model

Data processing and classification by means of AGA minimization suppose the formation of a data base, representing real objects by formal logic expressions. The initial task is to add a new object to the base. As it was shown above, in a general case, the structure of classes $\ell_j$ can be chosen arbitrarily, but as described above, AGA minimization will provide the correct regrouping and simplification of product terms, reducing it finally to the set of prime implicants. For simplicity, the given examples are given for visual 2D cases for low values of *k*.

- Algorithm to create the new object of class $\ell$, positioned in the cell $(\underline{x}_1,...\underline{x}_n)$.

1. Add the product term $\ell * X(\underline{x_1},\underline{x_1}) *...*((\underline{x_n},\underline{x_n})$ to the earlier obtained list of product terms. Note.Position in the list can be chosen intentionally in order to fasten the search of aggregated groups of terms.
2. If the added cell has neighboring cells with nonzero logic constants $\ell_j$, run the minimization procedure defined in Section 2.3.

As a whole, the work with classes will not differ greatly from the procedures for the search of prime implicants. Let's take for simplicity the minimized function $y = F(x_1, x_2)$, (see Figure 8a), defined only for eight truth levels $\{0,1,\ldots,7\}$. As in Section 2, it can be shown as a 2D grid containing 10 objects (white and black), which belong to classes $\ell = 2$ and $\ell = 5$.

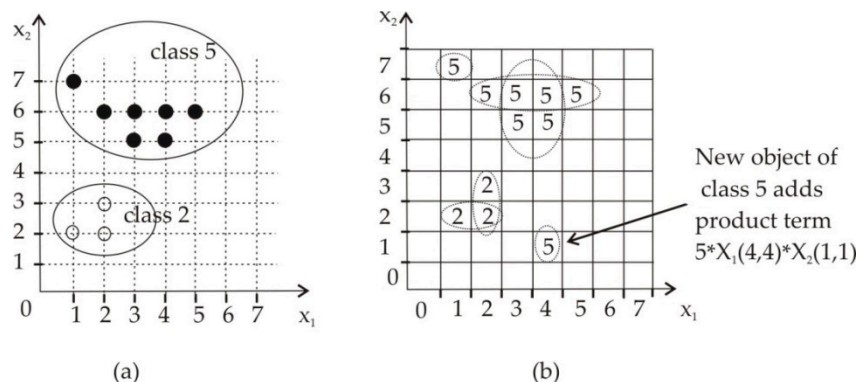

(a) (b)

**Figure 8.** (**a**) The example of the minimized MVL function of two variables, which is given as a 2D measurement grid including 10 objects classified by two classes $\ell = 2$, $\ell = 5$. (**b**) The structure of prime implicants of the function in class 5 is marked by six ellipses. If one adds a new object of class 5 to the cell (4,1) in the given structure, this event should be expressed by adding the new product term marked by the arrow.

This simple case responds to five prime implicants, which are shown by all possible variants to draw continuous horizontally or vertically oriented rectangulars composed from equal or greater logic constants in the grid. All these prime implicants are marked by ellipses and respond to the set of product terms

$$y = 5 * X_1(1,1) * X_2(7,7) + 5 * X_1(2,5) * X_2(6,6) + 5 * X_1(3,4) * X_2(5,6) +$$
$$+2 * X_1(1,2) * X_2(2,2) + 2 * X_1(2,2) * X_2(2,3). \tag{11}$$

Note that the object with coordinates (1,7) is given by class 5 and by the first prime implicant in expression (9); thus, it is an isolated object, as described by the separate rectangular segment of minimal size.

If the classifier given above is additionally learned by a new object of class 5, responding e.g., to point (4,1) of the grid, it should be described as the new product term $5 * X_1(4,4) * X_2(1,1)$, which is further to be added to expression (11). For such a simple example, one can easily see that the newly added object gives a separate product term, which will be the prime implicant if it is isolated from other objects.

- Deleting an object from the MVL classifier.

If the object is no longer relevant, its deletion procedure differs for minimized and not minimized MVL functions.

For the not minimized function, it is enough to substitute zero instead of the earlier given logic constant into the column $y$ of the truth table or into the appropriate matrix C of expression (4).

For the minimized MVL function, the object deleting is much more complicated, as it will need to destroy the set of prime implicants, containing the deleted object and to correct indexed pairs $(a_i, b_j)$ for all involved Literals.

- Algorithm to delete the object of class $\mathcal{C}$ from the AGA base.

The object to be deleted is positioned in the cell $(x_1, ..., x_n)$ and belongs to class $\mathcal{C}_j$.

1.  Find the aggregated group of all V cells, neighboring to the cell $(\mathbf{x_1, ..., x_n})$ and surrounded by DCSs. The set of constants used in the aggregated group is marked as $\mathcal{C}_j$, $j = 1,..,m$.
2.  Restore the set of product terms for individual cells in the aggregated group, according to Steps 3–5:

$$\mathcal{C}_j * X_1(a_{1,1}, b_{1,1}) * \ldots * X_n(a_{1,n}, b_{1,n})$$
$$\ldots$$
$$\mathcal{C}_j * X_1(a_{m,1}, b_{m,1}) * \ldots * X_n(a_{m,n}, b_{m,n}).$$

3.  Assign $j = 1$.
4.  For every input variable $x_1, \ldots, x_n$ in the jth product term $\mathcal{C}_j * X_1(a_{1,1}, a_{1,1}) * \ldots * X_n(a_{1,n}, b_{1,n})$ form the set of all possible Literals, as shown below:

$$\{X_1(a_{1,1}, a_{1,1}),\ X_1(a_{1,1}+1, a_{1,1}+1),\ X_1(a_{1,1}+2, a_{1,1}+2), \ldots,\ X_1(b_{1,1}, b_{1,1})\}$$
$$\ldots$$
$$\{X_n(a_{1,n}, a_{1,n}),\ X_n(a_{1,n}+1, a_{1,n}+1),\ X_n(a_{1,n}+2, a_{1,n}+2), \ldots,\ X_n(b_{1,n}, b_{1,n})\}$$

5.  Compose all possible combinations of product terms $\mathcal{C}_j * X_1(a_{1,1}, a_{1,1}) * X_2(a_{1,2}, a_{1,2}) \ldots * X_n(a_{1,n}, b_{1,n})$, containing the constant $\mathcal{C}_j$ and all possible combinations of Literals $X_1, \ldots, X_n$, obtained at Step 4:

$$\mathcal{C}_j * X_1(a_{1,1}, a_{1,1}) * X_2(a_{1,2}, a_{1,2}) \ldots * X_n(a_{1,n}, b_{1,n})$$
$$\mathcal{C}_j * X_1(a_{1,1}, a_{1,1}) * X_2(a_{1,2}+1, a_{1,2}+1) \ldots * X_n(a_{1,n}, b_{1,n})$$
$$\mathcal{C}_j * X_1(a_{1,1}, a_{1,1}) * X_2(a_{1,2}+2, a_{1,2}+2) \ldots * X_n(a_{1,n}, b_{1,n})$$
$$\ldots..$$
$$\mathcal{C}_j * X_1(b_{1,1}, b_{1,1}) * X_2(b_{1,2}, b_{1,2}) * \ldots * X_n(b_{1,n}, b_{1,n})$$

6.  Repeat Steps 4 and 5 for all other j.
7.  Run the minimization procedure.

The given above algorithm is opposite to the formation of an MVL function by means of the truth table and is very bulky. Respectively, Figure 9 visually demonstrates the corrections of the prime implicants set, which are necessary for the deletion of the object (Class 5) with grid coordinates (4,6) from the MVL function, as described above in Figure 8. The deletion procedure destroys the segment of Class 5 with two intersected prime implicants, given by the product term $5 * X_1(2,5) * X_2(6,6) + 5 * X_1(3,4) * X_2(5,6)$. It transforms it to the set of four prime implicants $5 * X_1(2,3) * X_2(6,6) + 5 * X_11(5,5) * X_2(6,6) + 5 * X_1(3,3) * X_2(5,6) + 5 * X_1(3,4) * X_2(5,5)$, which are marked by ellipses on the right in the Figure 9. Thus, for the large number of variables, the deletion of an object can be a tricky procedure, enlarging the computing time and disturbing clocking cycles. That is why often the adding and deletion of objects in the minimized MVL classifier motivates its use mainly for descriptions of static knowledge structures or in dynamic structures, as expressed by non-minimized product terms.

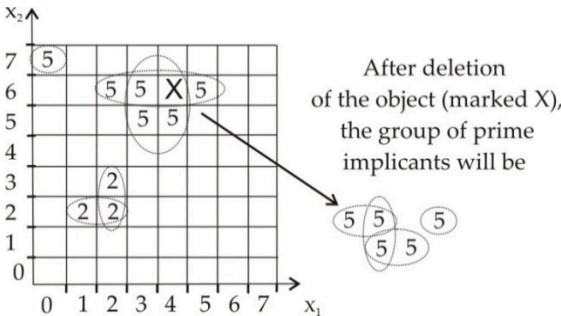

**Figure 9.** The deletion of the object from the minimized MVL function. The object to be deleted (marked X) belongs to Class 5. All the objects of Class 5 were initially given by two prime implicants, which are marked by ellipses. Such procedure enlarges the number of prime implicants in the localized group from two up to four, thus changing the computing time.

- The scheme to define formal expressions for a segment positioned near the separating hypersurface.

If there is some external formal method to calculate the border of the class of objects, then AGA can give a correct formal expression for the new segment, neighboring the border. The detailed discussion of this question is out of this paper, as it needs to discuss different space orientations for different segments of the border between classes. That is why only the general idea is shown here. Further, only the general idea is disclosed.

Let us consider the 2D function $y = F(x_1, x_2), k = 256$, shown in Figure 10 and defined as the product term

$$\Gamma_k = C_k * X_1(2,3) * X_2(7,7) + C_k * X_1(4,4) * X_2(5,6) + C_k * X_1(5,5) * X_2(3,4). \tag{12}$$

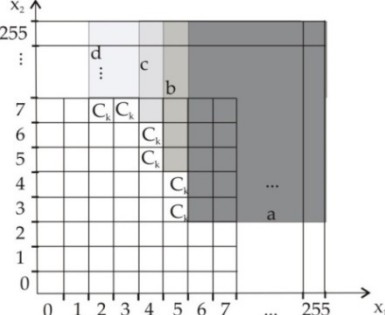

**Figure 10.** The principle scheme to define the formal expressions for a segment (marked gray), which is positioned near the separating hypersurface of class $C_k$.

This set of cells is regarded as the border of the class $C_k$, for which one should assign class $C_{k+}$ to all $x_1, x_2 > \Gamma_k$. In other words, it is necessary to obtain the formal expression for all cells, which are located to the right and up the sides of the $\Gamma_k$, which can be regarded as the separating curve. The appropriate logic function $S_{k+}$ can be constructed by the choice of the set of intersecting rectangulars, covering the given segment.

The choice of four rectangular segments (a), (b) (c) and (d) is clear from Figure 10. They should fill all the space between the curve (12) and the borders of the measurement grid, which have the maximal truth level 255 for all variables:

$$S_{k+} = C_{k+} * X_1(6, 255) * X_2(3, 255) + C_{k+} * X_1(5, 255) * X_2(5, 255) + C_{k+}$$
$$*X_1(4, 255) * X_2(7, 255) + C_{k+} * X_1(2, 255) * X_2(8, 255) \tag{13}$$

Thus, the possible scheme to describe the chosen segment $C_{k+}$ in the measurement grid is to modify the set of Literals in initial expression (12) as $X_i\,(a + 1, k - 1)$ for all the input variables and all cells, including the $\Gamma_k$. For example, the horizontal segment, consisting of two cells and given by the product term $C_k * X_1\,(2, 3) * X_2(7, 7),$ should be transformed to $C_{k+} * X_1\,(3, 255) * X_2(8, 255)$.

Expression (13) demonstrates the general idea of how one can further design the necessary set of operations that are necessary for AGA classifiers and the initial level of analysis of hypersurfaces in them.

What is substantial is that AGA logic expressions provide a correct model for any set of data, written either by prime implicants or non-minimized expressions. However, the problem here is to apply the appropriate set of methods to calculate the parameters of the border of the class, which cannot be solved directly by logic operators of AGA. It does not have embedded arithmetic operations and cannot directly calculate space parameters. That is why one should use appropriate software external modules in order to calculate MVL classification tasks.

### 3.4. The General Scheme to Analyze the MVL Classifier

As it was shown above, the modification of the MVL data set includes the search of isolated groups of cells formed by prime implicants with specific logic constants and surrounded by DCSs. It follows from the definitions in [71] that the general algorithm can be proposed to analyze the structure of classes in the learned MVL classifier. This method needs to graph the distribution of different values of input variables $x_1, \ldots, x_n$ for all logic constants $\{ C_1, \ldots, C_m \}$. It is shown in Figure 11 as the set of diagrams $C_1(x_1), \ldots, C_1(x_n), \ldots, C_m(x_1), \ldots, C_m(x_n)$, the overall number of which will be $m \times n$, where $m$ is number of used logic constants and $n$ is the number of input variables. The possible way to obtain correctly such diagrams is to subsequently scan all product terms in the MVL function. That potentially can be done just during its formation. For product terms with known logic constants $C_j\,\{ C_1, \ldots, C_m \}$, one should analyze all Literals $X_j(a, b)$ for all $1 \leq j \leq n$ and tag (by the vertical stroke in diagrams) the fact that $x_j \in [a, a + 1, \ldots, b]$, and thus this product term is not equal to zero. The algorithm can be proposed for the formation of such diagrams.

- Algorithm to obtain diagrams $C_1(x_1), \ldots, C_1(x_n), \ldots, C_m(x_1), \ldots, C_m(x_n)$.

Given parameters: $k-$ number of truth levels, $n-$ number of input variables, $\mathcal{C}_j-$ number of used classes, j = 1, .., m, where m $\leq$ k − 1.

1. Form the numbered (in arbitrary order) list of all product terms:

$$\mathcal{C}_j * X_1(a_{1,1}, b_{1,1}) * \ldots * X_n(a_{1,n}, b_{1,n})$$
$$\ldots$$
$$\mathcal{C}_j * X_1(a_{m,1}, b_{m,1}) * \ldots * X_n(a_{m,n}, b_{m,n}).$$

2. Define the set of arrays $C_1(x_1), \ldots, C_1(x_n), \ldots, C_m(x_1), \ldots, C_m(x_n),$ for $x_1, \ldots x_n \in L = \{0, 1, \ldots, k - 1\}$ and $C_1, \ldots, C_m \in L = \{0, 1\}$.
3. Assign $C_1, \ldots, C_m = 0$.

4. Assign

$$C_1(x_1) = 1 \quad \text{for } x_1 \in [a_{1,1}, a_{1,1} + 1, \ldots, b_{1,1}]$$
$$\ldots$$
$$C_1(x_n) = 1 \quad \text{for } x_n \in [a_{1,n}, a_{1,n} + 1, \ldots, b_{1,n}]$$
$$\ldots$$
$$C_m(x_1) = 1 \quad \text{for } x_1 \in [a_{m,1}, a_{m,1} + 1, \ldots, b_{m,1}]$$
$$\ldots$$
$$C_m(x_n) = 1 \quad \text{for } x_n \in [a_{m,n}, a_{m,n} + 1, \ldots, b_{m,n}].$$

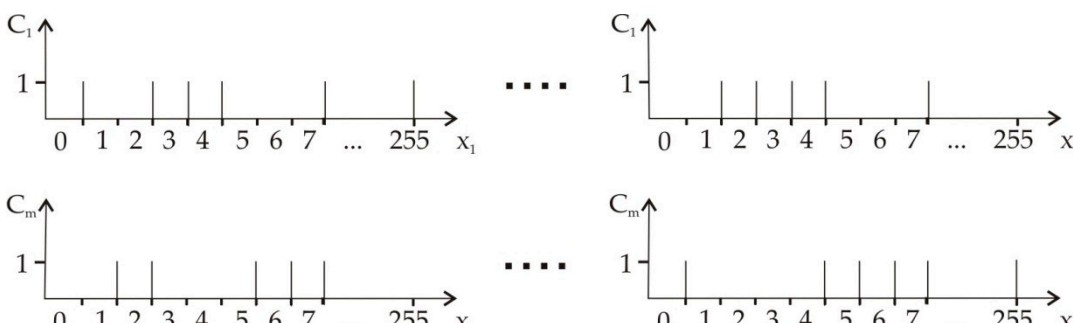

**Figure 11.** The set of diagrams $C_1(x_1), \ldots, C_1(x_n), \ldots, C_m(x_1), \ldots, C_m(x_n)$ is proposed to determine the repeatability frequency for different logic constants and to find zero gaps between aggregated groups of cells.

If one has earlier obtained some structure of prime implicants, namely that fact guarantees that diagrams will be filled non-uniformly and there will be some gaps near the borders of different classes. As prime implicants can be intersected [71], then the number of repeated values for $x_j$ obtained during the calculation diagrams does not matter. The method proposed above is cumbersome, but it is supposed to disclose the structure of any MVL function. Potentially, different hypotheses and estimates for needed corrections of the classifier can be based namely on the analysis of gaps in diagrams.

- The partial minimization for selected pairs of variables.

The given above examples of AGA-based procedures are illustrated by 2D grids in order to simplify the representation of the proposed method. However, for an MVL function with a large number of variables and classes, such analysis would be much more difficult, as one should deal with $n$-dimensional boxes. In order to process multiparametrical tasks, one may use additional steps in order to understand the possible structure of aggregated data. The minimization procedure [71] (described in Section 2) supposes the search and the deleting of product terms, which subsume other ones. This procedure involves the obligatory testing of all possible variables, which are taken for pairs of product terms and supplemented by the intentional choice of DCS. Principally, this method gives the possibility of running minimization firstly for some groups of variables and for some deliberately chosen DCS, and afterwards finishing this procedure for other ones. In this variant, one can initially perform minimization for pairs of space coordinates.

### 3.5. Heterogenious Architecture for the Integration of MAS

The growing complexity and variety of tasks for CN needs to work out the architecture, which is relevant to the integration of intellectual MAS robotics with quantum optics. The priority task is to form the structure of formal parameters for the description of the space of all the needed physical variables and to provide seamless computing for all types of robots. AI methods here include Boolean logic, fuzzy methods, and neural networks [27], which differ by basic logic models. MVL methods are mainly attractive for relevant description of high-dimensional spaces [71], as tree data structures [76] are actual for parallel computing. At the same time, quantum computing [44] is itself the way to

provide a new level of parallel computing. Thus, all these methods are interrelated, and the first step is to unite all these heterogeneous data processing methods within the general model, providing quick and clear navigation in the multi-parametrical space. Such a model should not only combine traditional IP protocols for addressing network nodes, but it should also provide targeted (or problem-oriented) addressing of AI algorithms, imitating human behavior. This goal demands modifying the architecture of network data processing.

The useful recent step to coordinate network data processing methods with AI technologies was done in QKD architectures [50,109,110]. QKD schemes employ complicated protocols, as they provide the generation of quantum keys and their application in classical cryptography coding [9]. As a result, QKD networks designers were forced to create new complicated architectures based on MAS. The integration of quantum optics with MAS model was considered in [50] for purely optical global photonic networks. This variant included the special level for protocol attributed to MAS and agents, which are specialized at the joint management of all quantum and traditional cryptography keys, which also can be interpreted as the attempt to minimize the access of the administrative staff to the processing of quantum cryptography keys [9]. The special key management center for human control of all cryptography resources was attributed to the highest level of hierarchical control, as it should protect the whole CN from all possible threats, and its work is specially regulated. The lower level was given to autonomous agents of special MAS, processing all the keys and isolating the network from the possible intervention of eavesdroppers or unloyal staff members.

A more specific version of network architecture was realized for the network-centric system with QKD lines, which was plugged via the standard fiber commutator to the trusted server [109,110]. The last scheme realized a trusted authority model for the "star" topology, which partially reproduced the agent's functions. As a result, this architecture also went beyond traditional ones.

Both of the above-mentioned quantum architectures with QKD lines are shown in Table 5 in comparison with a traditional open system interconnection (OSI) network model [87,111], which initially does not suppose any agents, robots, or quantum optics. OSI has not become the strictly supported standard, but it is the most visual means of disclosing the specific features of new schemes in comparison with the seven traditional levels of interaction of network protocols.

The highest level of priority for the architecture in [50] was given to the key management center for human control, which was attributed to the greatest ninth level in Table 4, as MAS responded to the lower eighth level, which responded to autonomous agents, processing all cryptographical keys. Such a structure seems to be adequate, as the "intellectual" and complexity level of key management agents is higher than ordinary applications, which is attributed to the seventh level of the OSI model. Any other intellectual agents with routine tasks should respond to additional levels between the seventh and the eighth ones. They can interact with application level 7, which coincides with the principle that OSI models mainly interact with either adjacent levels or via one level. Moreover, that scheme responds quite well to architectures [50,109,110].

The described above architecture is shown in Figure 12 and differs from other [50,109] network models, as it has the separate level of logic models, which should support in general three different types of logic models, which are namely traditional Boolean logic, fuzzy logic, and MVL. Here, neural networks are to be included into the Boolean logic type, as their threshold logic roots [59,60] are almost forgotten.

The MVL level of logic models is designed for targeted navigation and scheduling within the branched tree structure [76], describing all AI algorithms and robotic agents modules. MVL algorithms suppose the aggregation and processing of data and knowledge structures, which can be directly transmitted in peer-to-peer segments of MAS and integrated into CN. In other words, direct communications of robots and software agents should be provided by high-dimensional multi-parametrical space for the seamless addressing and target setting for robots. The simplest versions of such knowledge structures were earlier considered within the scheme of network mapping of MVL functions [76]. The task here is to provide the uniform model for the effective direct interaction of a software agent in the edge host computer with the distant hardware robotic agent.

**Table 5.** The difference between advanced quantum key distribution (QKD) architectures and a traditional OSI model.

| Levels of OSI Model | Standard OSI Model, (Key Words), [86,111] | Network-centric QKD Project, [109] | Photonics Networks QKD Project [50] |
|---|---|---|---|
| 9. | - | - | Key management center |
| 8. | - | - | Multiagent system for joint key management |
| 7. | Application (access to network services, HTTP, FTP, POP3, etc.) | Trusted authority ("Trent") was realized by the server | + |
| 6. | Presentation and secret coding (ASCII, EBCDIC) | + | + |
| 5. | Session (communication session control) | + | + |
| 4. | Transport (reliability of a link, segment, datagram, TCP, UDP) | + | + |
| 3. | Network (packet, routing and logical addressing, IPv4-6, IPSec) | + | + |
| 2. | Data link (bits, frames, physical addressing, Ethernet, network card, IEEE 802.22) | + | + |
| 1. | Physical (bits, cable, twisted pair, fiber optics, RF channel, USB ports,) | traditional means and QKD fiber lines | traditional means and QKD fiber lines and optical cross connects, directly plugged to switchers of IP routers |

The location of a separate level for fuzzy logic models is motivated by the specifics of approximate and uncertain models, which are needed primarily for actuators control, space positioning, and reasoning procedures. The description of algorithms for such subsystems suppose the large number of routine computing of fuzzy rules and membership functions [40–42]. The transfer of such knowledge structures does not need high-level programming and principally can be realized by FPGAs.

The location of a separate level for Boolean logic models is quite a natural step, as a specific feature of AGA modeling is its very limited tool kit, which initially does not include trivial arithmetic operations. The obvious solution here is the coordinated computing of conjugated algorithms, using a one-to-one mapping of the results obtained in AGA and Boolean logic modules. Another possible solution is that complicated functions can be defined as truth tables and logic functions, coordinated via network mapping schemes [76]. So, the higher level of MVL logic models in the proposed architecture does not diminish the role of Boolean logic and only reflects that different models are used for different tasks, and MVL is to provide additional secret coding by the MVL version of the OTP method [9].

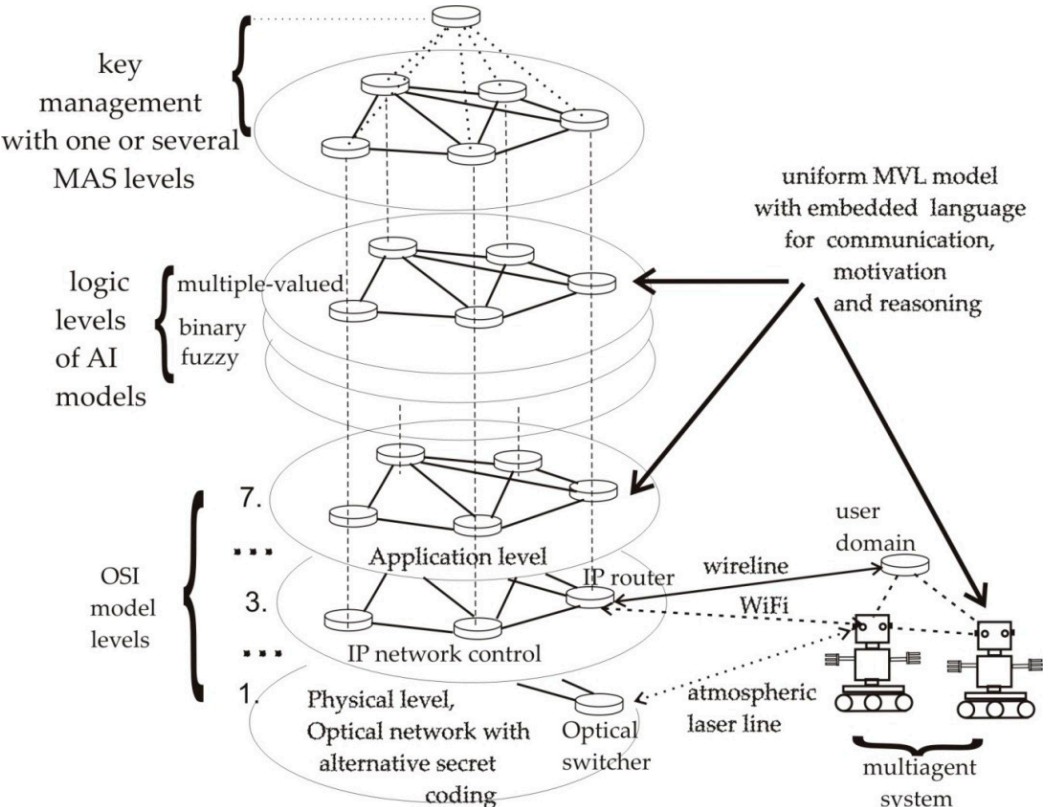

**Figure 12.** Proposed architecture for the integration of multiagent systems (MAS), traditional Boolean, and fuzzy logic by means of MVL functions.

### 3.6. The Structure of Knowledge and the Agent's Model, Necessary for Processing by AGA Methods

A substantial aspect of the proposed above heterogeneous architecture refers to the fact that intellectual agents in a network MAS are expected to provide many more intellectual algorithms and reduce traffic compared to applications in the traditional OSI model in Table 5. The difference between knowledge structures for modern and future robots is shown in Figure 13. Future agents are supposed to use computer vision and sensor networks for the autonomous evaluation of the scene parameters. Moreover, software and hardware agents in all nodes of CN need to use the same language for communications. That is why the united knowledge base structure for all robots in a MAS should be envisaged for peer-to-peer network communications.

Another aspect of the proposed above architecture is that both software and hardware robots should provide three main components, as shown in the Figure 14.

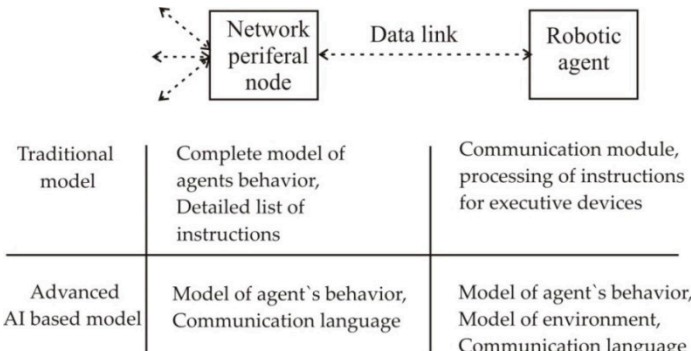

**Figure 13.** The difference between knowledge structures for modern and future versions of network agents.

The first component is to be composed from unchangeable and data written beforehand. For the AGA component of these data, the learning of the agent is considered as the targeted transfer of logic expressions, describing objects by the set of product terms, which supposes the united scheme of parameters for all variables $x_1, \ldots, x_n$ , constants $C_i$, and Literals $X(a, b)$. This component should imitate the intuitive decision making and can be compared to some extent with a pattern recognition system or a neural network.

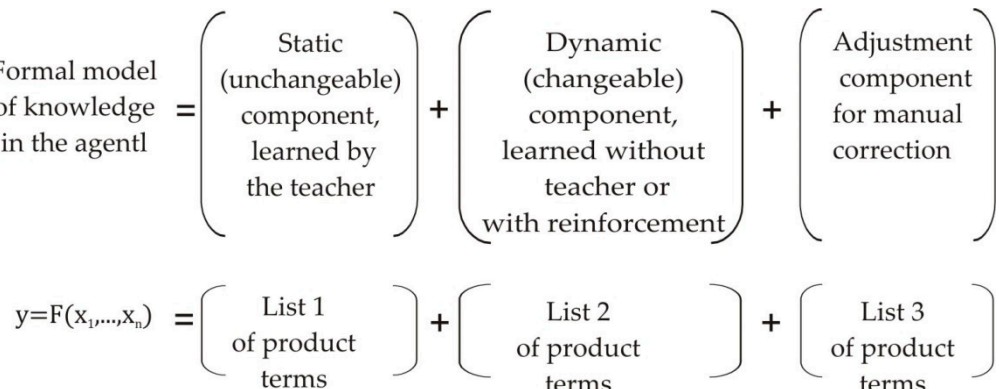

**Figure 14.** The model of knowledge for MVL representation.

The second component of the MVL knowledge structure is to be formed by different methods of robotic learning, which are attributed to the current scene of action. This dynamic component is preferably designated for not a large number of quickly appearing and disappearing objects of the scene, as it is principally difficult to modify a large-scale MVL model.

The third component for an MVL knowledge base should contain the set of all DCSs and completed corrections that are needed for debugging and testing.

## 4. Discussion of Possible Further Steps for Application of MVL Models

### 4.1. High Information Capacity MVL Model of a Language for Robotic Communications

The integration of peer-to-peer segments into a CN for the direct communication of robotic agents [50,51,69] should be supported by the special language or protocol, depicting vast data structures and complicated tasks. This problem seems to be actual not only for MAS, but also for IoT, sensor networks, intellectual cities, and houses.

The design of Nes C and Tiny DB languages, which are destined for large collectives of agents [65,66], has stimulated the design of an MVL-based language for simple robotic agents, and it is intended for the simplest microcontrollers, MCS-51 [68]. This MVL language protocol was proposed for instructions transmission via laser and twisted pairs. As the transmission of text instructions traditionally faces the problem of frequency analysis for different letters and words, the non-alphabet language for further secret coding was designed, using the transmission of homogeneous blocks of *k*-level logic values with a uniform distribution of repeatability frequency. To solve that task, the set of targeted vocabularies $W_1, \ldots, W_n$ should be preliminarily formed, where all vocabularies can contain *k* different words, and a transmitted phrase {w} should be formed by the strictly ordered sequence $\{w_1, \ldots, w_n\}$, which is composed of *n* *k*-valued numbers. For simple agents, $k = 256$ seems to be quite enough and responds to 8-bit hardware platforms [68]. The strictly ordered grammar construction "Message sender—Message receiver—Task (what should be done)—Where—When—Commentaries" consisted of eight words [64] and was adapted for the mass microcontroller MCS-51 with a limited number of memory registers. Such a model of the robotic language [68] supposes the transfer of a minimal set of bytes, which prevents the hidden penetration of malicious codes via service data.

The perception of MVL language phrases can be based on the scheme of the AGA classifier, using the MVL function with many input variables $y = f(x_1, \ldots, x_n)$. One can define the answering function $y = f(x_1, \ldots, x_n)$, where $x_1 \in W_1, \ldots, x_n \in W_n$, and $W_1, \ldots, W_n$ are the preliminary learned vocabularies, each containing k different words marked by k different truth levels. The task of the MVL classifier is to receive messages and produce responses. The number of different answers for the output variable y is limited by k values, but it can be drastically enlarged by the method considered in [80], where additional input variables $s_1, \ldots, s_\beta$ are used as scrolling parameters in $y = f_{answ}(x_1, \ldots, x_n, s_1, \ldots, s_\beta)$. These parameters are sequentually substituted by numbers from 1 up to $k - 1$ and instead of k, such a system will produce the enlarged number of responses.

The message or the query phrase in the robotic language can be generated also by the MVL function with the same structure $y = f_{quer}(x_1, \ldots, x_n, s_1, \ldots, s_\beta)$. One or several scrolling parameters will also provide the necessary length of the query.

The allocated level for MVL models, which was proposed in the architecture in Section 3.5, seems to be the necessary step to provide the joint work of MVL versions of OTP secret coding methods [67,80] with MVL language formal constructions, which are represented by the minimal number of bytes. Principally, for confidential communications within private MAS, an agent's messages can be additionally secretly coded by keys obtained from network QKD agents [9] or MVL modules for OTP secret coding [67]. Here, the allocated level of MVL models provides the chance to use simple schemes for the selective secret coding of robotic instructions and language constructions. For the separate level of MVL models, the owner of a hardware robot will receive the chance to choose additional cryptography means, supplementing the basic network cryptography facilities provided by network QKD agents. The user potentially can work out his own vocabularies, truth table, and language constructions.

Now, such a method can be supported by high-quality quantum random number generators [69,70], as well as by an MVL random oracle scheme and position-based cryptography algorithms [75,77,78].

## 4.2. MVL Emulation of Classification Models

The advantage of the complete set of AGA logic operators is the correct mathematical model, which is principally guaranteed for any finite set of input variables and truth levels. The switching function can be composed for arbitrary data, including both mathematical metrics and categorical data. The discussed above hierarchical structure of nested classes is good for formal symbolic parameters, where even an 8-bit platform provides up to 256 different classes.

The task of seamless computing supposes the detection of two kinds of "seams" in the multi-parametrical space by means of the AGA minimization procedure:

- the presence of voids in the truth table between adjacent blocks of rows in the truth table, if these blocks refer to different groups of data and there are gaps in the domain of the function,
- the overlap of blocks of rows in the truth table, if these blocks were obtained for partially matching training sequences or respond to different models.

The threat of the first situation is determined by the presence of undefined gaps in output signals domains, as the second situation leads to the redundant consumption of memory, energy, and computing time for a robotic agent. That is why the robotic component in the global CN requires the analysis of the learned knowledge base. A more complicated task is to compare software modules, which is needed for optimal knowledge base formation in a robot by means of network transfer and the exchange of software modules.

As quantum computing is in prospect a method to raise the network throughput, new logic methods should be able to use it, too. The discussed above method of MVL minimization is based on a sequential search of the subsuming product terms and prime implicants. In practice, it will be necessary to scan a large number of rows and columns of arrays (4), containing logic constants and parameters of Literals. The idea to analyze the content of vast data structures by means of AGA demands a large volume of computing. As the quantum Grover algorithm [112] is now regarded as a well-investigated algorithm for data search in quantum networks, potentially it can be used also for AGA minimization. The detailed discussion of this question is out of the frames of the presented work, but AGA minimization seems not to create substantial obstacles to apply the Grover algorithm.

## 5. Conclusions

Future generations of CN demonstrate the trend for integration of quantum technologies and AI methods, because modern global networks are to control autonomous robotic and collective multiagent systems. Another motivation is to use QKD methods for reliable a one-time pad secret coding of traffic and to protect future distributed quantum network computing. One more reason is the necessity of preventing data leakage and controlling non-loyal staff by means of AI technologies.

The earlier designed methods and schemes of multiple-valued logic, which are intended for secure data coding in small-scale microcontroller systems, also have revealed the need to design AI algorithms for robotic multiagent systems. Attempts to model multiple-valued functions for secret coding and different agents' subsystems demonstrate the need to use cumbersome logic minimization procedures, which in practice suppose the solution of multi-criteria optimization tasks and also motivate the use of AI methods.

The presented paper discusses the possible scheme for MVL-based data clustering, which obtains a hierarchical and nested structure of classes for arbitrary data that are characterized only by different classes. This method does not suppose any preliminary given hypothesis or estimates of data parameters based on metrics or statistical parameters. Such a method is a candidate for actual mixed clustering, combining numerical and categorical values.

The MVL minimization clustering scheme can be supplemented by the correct formation of an MVL model for arbitrarily given borders of classes and by MVL schemes for adding/deleting objects and for a detailed description of segments near the borders.

In order to coordinate an MVL algorithm with the well-known precise and approximate fuzzy logic metrics employed in data clustering, the heterogeneous architecture is proposed with allocated levels for Boolean, fuzzy, and MVL logic models. Such a choice is dictated by the necessity of providing additional MVL-based secret coding for robotic agents. Another reason is the need to coordinate a large number of various logic models and metrics, in order to form the complicated logic model of an agent.

Further research of the MVL agent's model and the proposed architecture is supposed to be continued by the design of the special language with high information capacity, which can provide seamless computing on platforms with different resources. Another task for it is the correct representation of models containing mixed data. Full-scale MVL modeling potentially can be simplified by the quantum Grover algorithm, which is used for the search of subsuming product terms.

**Supplementary Materials:** The following are available online at http://www.mdpi.com/2624-960X/2/1/10/s1, the version of a program is given in supplementary materials, which was developed in the C language for the debugging and testing of microassembler programs, providing MVL algorithms in microcontrollers MCS-51.

**Funding:** The presented paper received no external funding.

**Conflicts of Interest:** The author declares no conflict of interests with other persons and groups.

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
