# Peer review of "Heterogeneous Network Architecture for Integration of AI and Quantum Optics by Means of Multiple-Valued Logic"

_quantumrep, doi:10.3390/quantum2010010_

Round 1

Reviewer 1 Report

In this paper, the authors have studied "Heterogeneous network architecture for integration 3 of AI and quantum optics by means of 4 multiple-valued logic". After reviewing the whole paper I like the physics content of the paper. I am accepting this paper as it is (without any modification) for the publication in Quantum Reports.

Author Response

The author is grateful to all reviewers for useful critical remarks.

The revised version of the paper included:

The review of traditional (sec.1.6) and fuzzy data clusterization (sec.1.7) methods were added. The review (sec 1.3) of data security methods for MAS was added. The aim of the paper and conclusions were formulated more clearly and closer to MVL classification , less important discussions were excluded. The structure of the work was changed in order to disclose the need for special architecture. The version of C program for the calculation MVL function was added in Appendix. The minimization method and prime implicants formation was represented in detail. Algorithms for MVL classification were added. A pair of illustrative examples were shortened. The discussion of possibility of MVL modeling of a neural network was excluded in order to prepare it more thoroughly for a separate paper.

Reviewer 2 Report

This manuscript proposes a heterogeneous architecture for integration of AI and quantum optics. The main problem is that quantum optics often works in the low temperature limit. It is well known that the numerical verification by master equation is needed in this regime. So, I doubt that the manuscript presents new contribution to warrant publication.

Author Response

The author is grateful to all reviewers for useful critical remarks.

The revised version of the paper included:

The review of traditional (sec.1.6) and fuzzy data clusterization (sec.1.7) methods were added. The review (sec 1.3) of data security methods for MAS was added. The aim of the paper and conclusions were formulated more clearly and closer to MVL classification , less important discussions were excluded. The structure of the work was changed in order to disclose the need for special architecture The version of C program for the calculation MVL function was added in Appendix. The minimization method and prime implicants formation was represented in detail. Algorithms for MVL classification were added A pair of illustrative examples were shortened The brief discussion of MVL modeling of a neural network was excluded in order to prepare it more thoroughly for a separate paper.

Reviewer 3 Report

To enhance the performance of network, the authors propose a heterogeneous network architecture with three levels of AI logic modeling in order to simplify the control of network robotic systems. Multiple-valued logic is regarded as the platform to extend network addressing space by high dimensional space for targeted control of various AI algorithms, including multi-agent systems, neural networks and fuzzy logic. In my opinion, this work is meaningful for AI network. There are some comments below which I recommend to give one chance to take a revision. A more comprehensive literature survey may be provided. For instance, the authors missed the following related works. Resource Assignment based on Dynamic Fuzzy Clustering in Elastic Optical Networks with Multi-core Fibers, IEEE Transactions on Communications. The setting of simulation should be descripted in more details.

Author Response

The author is grateful to all reviewers for useful critical remarks.

The revised version of the paper included:

The review of traditional (sec.1.6) and fuzzy data clusterization (sec.1.7) methods were added. The review (sec 1.3) of data security methods for MAS was added. The aim of the paper and conclusions were formulated more clearly and closer to MVL classification , less important discussions were excluded. The structure of the work was changed in order to disclose the need for special architecture. The version of C program for the calculation MVL function was added in Appendix. The minimization method and prime implicants formation was represented in detail. Algorithms for MVL classification were added. A pair of illustrative examples were shortened. The brief discussion of MVL modeling of a neural network was excluded in order to prepare it more thoroughly for a separate paper.

Round 2

Reviewer 2 Report

The authors have made a substantial revision and I think this paper may be publishable.